# The Aerosol Limb Imager: Multi-spectral Polarimetric Observations of Stratospheric Aerosol

Daniel Letros<sup>1</sup>, Liam Graham<sup>1</sup>, Adam Bourassa<sup>1</sup>, Doug Degenstein<sup>1</sup>, Paul Loewen<sup>1</sup>, Landon Rieger<sup>1</sup>, and Nick Lloyd<sup>1</sup>

<sup>1</sup>Department of Physics & Engineering Physics, University of Saskatchewan, 116 Science Place, Saskatoon, Saskatchewan S7N 5E2, Canada

**Correspondence:** Daniel Letros (daniel.letros@usask.ca)

**Abstract.** The Aerosol Limb Imager (ALI) is designed to measure stratospheric aerosol by imaging limb-scattered sunlight. Each image taken by ALI is spectrally filtered at a tunable wavelength, and refined to consist of either horizontally or vertically polarized light. Novel to limb imaging, these polarized observations of ALI provide a means to isolate tangent altitudes which have signal contaminated by clouds as identified by the depolarization of the scattered radiance. This avoids the ambiguity caused by clouds to be interpreted as aerosol in a retrieval. We present a polarized aerosol retrieval methodology which retrieves vertically resolved aerosol number density, and median radius of a unimodal log-normal distribution, in addition to a scalar (uniform in altitude) width of the log-normal distribution. We explore the cloud discrimination and aerosol retrieval of ALI in simulation as validation of the efficacy and the limits to the technique. We then apply the retrieval to three example sets of observations taken from the most recent high-altitude balloon flight of ALI. One set provides a nominal exemplar, while the other two represent more difficult retrieval conditions of an increasingly polarized atmosphere. We compare the aerosol extinction of ALI in all three exemplar cases to the best coincident extinctions of three space based instruments: the Stratospheric Aerosol and Gas Experiment (SAGE III/ISS), the Ozone Mapping and Profiler Suite Limb Profiler (OMPS-LP), and the Optical Spectrograph and InfraRed Imaging System (OSIRIS). We provide discussion to the agreement of all three cases against the comparison instruments with respect to the efficacy of our approach. However, we find the retrieved aerosol extinction of ALI in the nominal case (considering the limitations of ballooning) is in good agreement (a median absolute percent difference 

**Figure 1.** Render of ALI optical layout. The front end optics consists of a baffle, stops, and two off-axis parabolic mirrors providing angular magnification. The spectral and polarization selection (SPS) sub-system is comprised of four components (listed front to back): the Liquid Crystal Rotator (LCR), a vertically orientated linear polarizer, the AOTF, and a horizontally orientated linear polarizer. The back end optics comprises one off-axis parabolic mirror focusing a spectrally filtered and polarized image onto the detector.

tuned frequencies (center filter wavelengths). The diffraction efficiency at a tuned wavelength of the AOTF is taken as the peak percentage difference between the diffracted and non-diffracted signals. The resolution of the spectral bandpass of ALI at a tuned frequency is taken as the full-width half-max (FWHM) of the diffraction response. Figure 2 shows this process and results over the operating range of the AOTF.

We must note that to avoid very significant computational requirements of modelling a high-resolution spectrum for the aerosol retrieval, we treat all measured photons at a tuned wavelength to be of that tuned wavelength. Effectively this ignores the change of wavelength within the resolution of a spectral response. To account for the net photons ALI measures at each tuned wavelength, the area of each spectral response is calculated by integrating the measured diffraction responses in Fig. 2(b) with respect to wavelength after they have been normalized by the diffraction efficiency. This area yields a scalar factor for each tuned wavelength that is used to account for the wavelength dependence of radiance as the images are processed.

# 2.2 Polarimetric Response


As mentioned before, ALI is designed to image the vertically polarized limb-scattered sunlight of the atmosphere, or the horizontally polarized light of the atmosphere depending on the configuration of the LCR. However to be more precise, we consider all polarimetric behaviour in terms of the Stokes parameters I, Q, U, and V (following their typical definitions (Bass et al., 2010)) and the Mueller matrices which transform them. Each pixel of an ALI image measures I', which is produced by transforming the atmospheric Stokes vector of that pixel's line of sight by the Mueller matrix of ALI:

**Figure 2.** AOTF spectral performance. (a) Measurement process of AOTF spectral response at two tuned frequencies. These frequencies correspond to shifting diffraction to a desired wavelength. This diffracted light is imaged by ALI. 82 MHz (1058 nm) shown in blue and 80 MHz (1083 nm) shown in orange. The total un-diffracted light with no active diffraction is shown in black. (b) Measurements of plot (a) shown as two spectral responses. The diffraction efficiency (DE), i.e. the AOTF transmission, is calculated by peak response, and spectral resolution is defined as the FWHM of the response. (c) Diffraction efficiencies of the AOTF sampled across the full operating range of the AOTF as a function of diffracted wavelength. (d) The spectral resolution (FWHM of (b)) for the full range of the AOTF as a function of diffracted wavelength.

$$I' = \begin{bmatrix} 1 & 0 & 0 & 0 \end{bmatrix} \mathbf{M_{ALI}} \begin{bmatrix} I_{\text{atmo}} & Q_{\text{atmo}} & U_{\text{atmo}} & V_{\text{atmo}} \end{bmatrix}^T$$
 (1)

where  $M_{ALI}$  is an appropriate Mueller matrix of ALI for the measurement. Since ALI has two LCR states defining the polarimetric behaviour, ALI effectively has two wavelength dependent Mueller matrices - one matrix for when the LCR is toggled on, and another for when it is toggled off. The work of (Letros et al., 2024) describes the polarization characterization we apply to ALI, and this procedure produces the full 16 element Mueller matrix at the required states of interest. Therefore we do not discuss this topic with depth here.


However to provide context to discussion in the present work, when the LCR is enabled (referred to as the "LCR on" state) ALI behaves as an imperfect vertical linear polarizer. Likewise, when the LCR is not enabled (referred to as the "LCR off" state) ALI acts as an imperfect horizontal linear polarizer. This non-ideal response of ALI (in the Stokes basis defined by ALI) as found in (Letros et al., 2024) is shown in Fig. 3, and implications of this response to the analysis of atmospheric polarization is discussed in Section 4.2.4.

Figure 3. Non-ideal polarimetric response of ALI (Letros et al., 2024) is shown as normalized coefficients of the first row of the Muller matrix, which dictates the measured Stokes parameter I'. Plot (a) is the response of the LCR on state. Plot (b) is the response of the LCR off state. The coefficient of  $m_{01}/m_{00}$  shows the proportional transference of  $Q_{\text{atmo}}$  into I'. Likewise,  $m_{02}/m_{00}$  and  $m_{03}/m_{00}$  show the same for  $U_{\text{atmo}}$  and  $V_{\text{atmo}}$  respectively. An ideal ALI response would have  $m_{01}/m_{00} = -1$  in the LCR on state,  $m_{01}/m_{00} = 1$  in the LCR off state, as well as  $m_{02}/m_{00} = 0$  and  $m_{03}/m_{00} = 0$  for both cases - i.e. purely vertical linear polarization with the LCR on, and purely horizontal linear polarization with the LCR off.

Each pixel in an ALI image measures a signal reported as the raw detector units of DN, and we convert this measurement to

# 2.3 Image Correction




the more meaningful measure of radiance in units of photons/s/cm²/sr/nm. In this conversion, we also handle correction of instrumentation effects such as dark current, photo response non-uniformity, optical flat fielding, and bad pixels. The resulting tool of this process is a database of pixel-by-pixel coefficients which can synthetically reproduce the ALI measurement of a calibrated broadband integrating sphere of known (randomly polarized) spectrum. These synthetic images can be made for all wavelengths, LCR states, and exposure times of interest. Furthermore, since they reproduce the spatially homogeneous and full-field conditions of the integrating sphere, they can be used to relate the ALI measurements to the external source for each pixel measured. We provide a summary overview to the construction and application of this database within scope of this paper.

The database of coefficients begins by characterizing the dark behaviour of the ALI detector. As typical, we collect a large set of images (hundreds) at various exposure times when no light is present within the optics. While the focal plane array of the detector is thermally controlled by a thermoelectric cooler, the temperature of the other electronics in the detector have an impact on the observed dark signal. We analyse the dark image set for this dependence and fit an exponential form to each pixel quantifying this behaviour. All pixels are then corrected to a common electronic temperature according to these fitted curves. The temperature corrected images are then used to determine the expected dark signal each pixel is expected to produce for a given exposure time. This is quantified by another regression with respect to exposure time, and is mostly linear. However,

some non-linear behaviour is observed for short exposure times which we capture with another exponential regression. Note

**Figure 4.** Example comparison of synthetic image construction for ALI correction. (a) A single ALI image of the calibrated integrating sphere. AOTF is tuned to diffract 1450 nm, the LCR is off, and an exposure time of 0.450 seconds is used in this example. Dark correction has been applied. White dots in this image indicate the bad pixels as determined only by the dark correction. (b) A synthetic reproduction of the real image shown in the left plot constructed from the database of calibration coefficients. White (bad) pixels seen in this image are determined as pixels with non-ideal responses in both dark and illuminated conditions. These pixels are discarded. (c) A histogram of all (non-bad) pixel values after the synthetic (middle) image is subtracted from the real (left) image

that any detector pixels which fail to regress well and/or are statistical outliers are marked as bad pixels in the dark calibration. These pixels are not carried forward in image processing.




Following this, a large collection of images is taken of the calibrated integrating sphere mentioned before at a fixed intensity. This set of images consists of various exposure times at each tuned wavelength of the AOTF, and both toggled states (polarizations) of the LCR. All images are corrected for the dark behaviour as discussed above, and a linear regression is applied which quantifies each pixels response with respect to exposure time in each configuration (AOTF tuned frequency and LCR state) of ALI. This fitting can then be used to construct an expected ALI image given a homogeneous source after dark correction. Similar to the bad pixel identification of the dark regression, any pixel which fails under illumination is also marked as bad and not carried forward in image processing.

Figure 4 shows an example of synthetic image construction compared against an actual ALI image. As this example shows, the synthetic image provides a faithful recreation of the actual ALI measurement. The error of this reproduction, as demonstrated by the histogram FWHM of 33 DN, is approximately 0.4% of the mean of the image signal.

Since the source of this exercise is known and spatially homogeneous, these synthetic images effectively provide the corrections of photo response non-uniformity and optical flat fielding. In addition, the source spectrum of the calibrated integration sphere is known. Therefore, we can also provide an absolute calibration by relating the detector counts to the source radiance. This is done by taking the spectrum of radiance used in the calibration and integrating it over the AOTF bandpass to determine a unit conversion from DN into  $photons/s/cm^2/sr$  (radiance without the wavelength dependence). The wavelength dependence is introduced back into this conversion in accordance with the discussion in Section 2.1 to ultimately give radiance

in photons/s/cm<sup>2</sup>/sr/nm. The uncertainty on each pixel is then calculated as a function of shot noise of the corrected image, dark shot noise at the same exposure time, and detector read out noise.

A final concern we address in image correction is the issue of (potential) stray light. A unique advantage of AOTF technology is that if an image is taken where there is no diffraction, then that image is a measure of the stray light in the optical system - and one that is applicable to the illumination conditions outside of the instrument when the desired measurements (AOTF diffraction on) is being taken. The image acquisition strategy of ALI is to take an image without diffraction (AOTF off) for every image with diffraction (AOTF on). The AOTF off images are corrected in the same manner as the AOTF on images to produce an image of stray light. This stray light image can then be subtracted off the AOTF on image. Note that this method effectively deals with internally scattered stray light but does not handle the impact of out-of-field stray light for a given scientific image. This is mitigated with careful baffling of the input aperture.

# 3 Flight Campaign and Spectral Results




The ALI measurements we use in the aerosol retrievals of the present work are taken from the most recent high-altitude balloon flight of ALI. This flight began on August 21st, 2022, 11:30 pm (local) out of the Timmins stratospheric balloon base (attached to the Victor M. Power airport) in Ontario, Canada. ALI was situated on the balloon gondola and orientated such that when the gondola is flat and level, the highest lines of sight (top pixels of the ALI detector) would be horizontal and with tangent locations on the instrument itself. ALI ascended to a float altitude of  $\geq 35$  km roughly two hours after the 11:30 pm launch. However, being a night launch ALI only began useful observations as the sun was rising. At this point in the flight the gondola was steered to maintain a solar azimuth angle (SAA) of 60°. During the flight the gondola position and orientation is recorded which allows reconstruction of all the ALI lines of sight in each image.

During the flight, ALI took images in sets which make up an ALI science scan. A full science scan consisted of imaging 710 nm, 750 nm, 805 nm, 865 nm, 985 nm, 1025 nm, 1090 nm, 1105 nm, 1230 nm, and 1450 nm. At each wavelength an image with the LCR off (horizontal polarization) is taken, then AOTF off, then LCR on (vertical polarization), and AOTF off again. The LCR on and LCR off images have the AOTF on to image the atmospheric limb, while the AOTF off imaging provides stray light correction.

A total of 153 scans were collected during the flight, but for the scope of the present work we select only three individual science scans for demonstration. A main reason for this is the nature of the balloon attitude and flight path, which results in many of these scans essentially measuring the same atmospheric scene as others taken close in time. This reduces the number of effective atmospheric measurements within this context. Furthermore, we consider that the three selected scans present conditions which make for robust discussion to our approach (and limitations). Additionally, for each of the three scans we retrieve aerosol using only 750, 1025, and 1230 nm. This is to avoid spectral contamination from trace gas absorption in the wings of the AOTF passband at the other channels. These scans are compared to observations of SAGE III/ISS, OMPS-LP, and OSIRS in Section 5. Information of these external observations and their coincidence to ALI observations is provided in the supplemental document.

Table 1. Science Scan Summary


| Scan        | UTC [2022-08-22] | Lat / Lon [deg] | SAA / SZA / SSA [deg] | Effective albedo | Approximate DoP |
|-------------|------------------|-----------------|-----------------------|------------------|-----------------|
| 1 (nominal) | 14:06:13         | 45.87 / -82.24  | 60.0 / 56.0 / 65.2    | 0.833            | 25% - 30%       |
| 2           | 10:30:26         | 51.61 / -81.71  | 60.1 / 90.8 / 60.2    | 0.3              | 40% - 50%       |
| 3           | 15:19:13         | 46.28 / -83.84  | 59.9 / 46.2 / 68.4    | 0.615            | 50% - 80%       |

As the gondola was aloft during the near 13 hours of flight, sequences of images were taken and categorized into different scans. Three of these scans are selected for study in this work and their metrics are summarized here. Latitude and longitude indicate the tangent position of the ALI scans. Solar azimuth angle (SAA), solar zenith angle (SZA), and solar scattering angle (SSA) are all given with respect to the ALI view. All of these properties are reported as the average of the complete scan. Discussion of effective albedo is deferred Section 4.1 but shown here for convenient summery. The approximate DoP is also given as a convenient summery of the atmospheric polarization found in Fig. 14. Information indicating the coincidence of these scans with observations of SAGE III/ISS, OMPS-LP, and OSIRS are provided in the supplemental document.

The first scan we select is of reasonably nominal conditions for ALI to observe. The balloon gondola was relatively stable for this scan compared to most others taken during the flight (gondola attitude within IMU error during exposures), and ALI observes an obvious cloud layer in the lower portion of these images with clear sky perceived above. The other two scans we select present more difficult observation conditions both in terms of relative gondola stability (although there was still minimal attitude change during exposures, i.e  $

**Figure 5.** Observations of all three ALI scans used in the present work. Panels (a, b, c) are Scan 1 observations, (d, e, f) are Scan 2 observations, and (g, h, i) are Scan 3 observations. Panels (a, d, g) are example images from the three different scans taken at 1230 nm with the LCR off. Panels (b, e, h) are radiance profiles of different wavelengths with the LCR off (horizontally dominate polarization), and panels (c, f, i) are radiance profiles of different wavelengths with the LCR on (vertically dominate polarization).

# 4 Retrieval Methodology and Prototyping

Here we discuss the concepts behind the inversion methodology used by the present work, as well as also providing results of prototyping the algorithms in simulation with known true states. This not only demonstrates the efficacy of the retrieval algorithms we present here, but also contextualizes the results we obtain when applying the algorithms to the real data of the flight in Section 5. As mentioned before, the aerosol profiling ability of ALI has two main aspects we discuss here. The first aspect is determining altitudes of cloud contamination using retrieved DoP profiles of the atmosphere. These altitudes are then passed on to the second aspect of the (separate) aerosol retrieval and mark the lower altitude limit to retrieve. Facilitating both

the DoP and aerosol retrievals is the need to estimate an effective albedo factor to use in the forward modelling, this will also be briefly discussed.






Both aerosol and DoP retrievals have separate implementations of the same underlying inversion theory, which is based in the standard approach of (Rodgers, 2000). This approach attempts to find the statistically most likely state vector  $\boldsymbol{x}$  given the observations encapsulated in measurement vector  $\boldsymbol{y}$ , under the assumption of normally distributed probability density functions. For non-linear systems, this is iteratively done using Equation 5.35 of (Rodgers, 2000):

$$\boldsymbol{x}_{i+1} = \boldsymbol{x}_i + (\mathbf{S}_a^{-1} + \mathbf{K}_i^T \mathbf{S}_{\epsilon}^{-1} \mathbf{K}_i + \gamma \mathbf{D}_n)^{-1} \{ \mathbf{K}_i^T \mathbf{S}_{\epsilon}^{-1} [\boldsymbol{y} - \mathbf{F}(\boldsymbol{x}_i)] - \mathbf{S}_a^{-1} [\boldsymbol{x}_i - \boldsymbol{x}_a] \}$$
(2)

where i notes the iteration, the a-priori state vector  $\mathbf{x}_a$  encapsulates the a-priori knowledge of the state,  $\mathbf{K}$  is a Jacobian,  $\mathbf{S}_a$  is the covariance of  $\mathbf{x}_a$ , and  $\mathbf{S}_\epsilon$  is the covariance of  $\mathbf{y}$ .  $\mathbf{D}$  is a customizable Levenberg-Marquard dampening matrix which retards the change in the state vector from one iteration to the next. The scalar  $\gamma$  controls the dampening strength of  $\mathbf{D}$  and is adjusted to be larger or smaller based on the change of the underlying cost function that is being minimized. Ideally,  $\gamma \to 0$  as the retrieval progresses. An additional matrix, the averaging kernel, is defined as  $\mathbf{A} = (\mathbf{S}_a^{-1} + \mathbf{K}_i^T \mathbf{S}_\epsilon^{-1} \mathbf{K}_i)^{-1} \mathbf{K}_i^T \mathbf{S}_\epsilon^{-1} \mathbf{K}$  and yields useful metrics about the retrieval - such as information content, and vertical resolution.

The remaining element of Equation 2 for definition is  $\mathbf{F}(x_i)$ . This is the forward model of the inversion which models the same kind of observations within y that would be produced by the given state  $x_i$ . The forward modelling of the present work is done with the radiative transfer model SASKTRAN (Bourassa et al., 2008; Zawada et al., 2015) coupled with an ALI simulator following the heritage of (Kozun et al., 2020). Briefly speaking, SASKTRAN calculates the expected atmospheric Stokes vectors, as well the polarized Jacobian  $\mathbf{K}$ , given some atmospheric state and observational geometry. The instrument simulator adjusts the Stokes basis to account for the attitude of the gondola, and then applies the appropriate ALI Mueller matrix to model the LCR on or off observation. The only significant exception to this forward modelling dynamic is in the DoP retrievals where SASKTRAN is only used to produce a-priori information, but this is discussed in Section 4.2.

During our retrievals, Equation 2 is run until convergence is determined. This is evaluated based on an established method of evaluating the cost function  $\chi^2$  (Rodgers, 2000; Zawada et al., 2018):

$$\chi^2 = [\mathbf{F}(\boldsymbol{x}) - \boldsymbol{y}]^T \mathbf{S}_{\epsilon}^{-1} [\mathbf{F}(\boldsymbol{x}) - \boldsymbol{y}] + [\boldsymbol{x}_{\boldsymbol{a}} - \boldsymbol{x}]^T \mathbf{S}_{a}^{-1} [\boldsymbol{x}_{\boldsymbol{a}} - \boldsymbol{x}]$$
(3)

for both a non-linear and linear iteration. The ratio of these two  $\chi^2$  values is taken, and if this ratio is one within a specified tolerance - taken as 0.001 in the present work - then it is an indication the linear estimate is now as good as the non-linear estimate and a solution has been reached. At this point, the uncertainty of the final state estimation  $\hat{x}$  can be determined by the solution covariance matrix  $\hat{S}$  (Equation 5.13 (Rodgers, 2000))

$$\hat{\mathbf{S}} = (\hat{\mathbf{K}}^T \mathbf{S}_c^{-1} \hat{\mathbf{K}} + \mathbf{S}_c^{-1})^{-1} \tag{4}$$

There are two potentially notable deviations we make from the common approaches of this inversion technique. The first is that typically for atmospheric inversions one will implement a Twomey-Tikhonov regularization matrix (Rodgers, 2000; Twomey, 1963; Tikhonov, 1963). We do not adopt this approach as we found it harmed vertical resolution of our retrievals more than improving the smoothness of the state vectors. We simply specify a-priori uncertainties as discussed in Section 4.3. Second, our retrievals use measurement and state vectors of large dynamic range which tends to produce ill-conditioned inversions. Since the inversion technique of (Rodgers, 2000) is a variant of the Extended Kalman Filter (Kalman, 1960; McGee et al., 1985; Becker, 2023; Grewal, 1993), we adopt the Singular Value Decomposition - Kalman Filter (Wang et al., 1992; Kulikova and Tsyganova, 2017) to combat this. In brief, this method uses singular value decomposition to enforce positive definite matrices and largely propagates the inversion in eigenvector space.

#### 4.1 Albedo Estimation






The albedo estimation aims to find the albedo to use in the radiative transfer forward model which best matches the observed radiance at high-altitudes over the available spectrum of ALI in each scan. This is a typical step for limb scatter aerosol retrievals (Bourassa et al., 2012). Once this is determined we consider it fixed for the forward modelling purposes of both the DoP retrieval and the aerosol retrieval. While the albedo could be included as a parameter of the aerosol retrieval proper, we take this ad-hoc approach to constrain and simplify the aerosol retrieval forward modelling rather than to include it as a member of the state vector x to be optimized. In addition, high-altitude normalization of the aerosol retrieval's measurement vectors is a strategy to mitigate systematics and forward modelling error such as improper albedo (Rieger et al., 2018).

For purposes of the albedo estimation, we define high-altitude as all observed tangent altitudes between 33 km and 34 km in the science scan. It is expected that at these altitudes the influence of aerosol is minimal. Therefore, adjusting albedo in a forward modelled Rayleigh (no aerosol) scattering atmosphere to match the observed high-altitude signal will provide an *effective* albedo accounting for the up-welling radiation. The measure we take to quantify the albedo is the integration of the high-altitude radiance with respect to wavelength. A flow diagram outlining this algorithm is provided in Fig. 6 and discussed here.

To begin, high-altitude radiance at different configurations (LCR state and wavelength) are collected and a mean radiance value between 33 km and 34 km is found for each configuration (Fig. 6(a)). These values have measurement noise, and for this reason these values as a function of wavelength will vary (represented with the dotted line in Fig. 6(b)) about the expected background trend. The noisy data is used in regression to a decaying exponential representing the trend in the data (solid line of Fig. 6(b)). This curve is then integrated producing a spectral area as the metric for the albedo estimation. The ALI configurations are then forward modelled (without noise) for both a high (beginning at 1.0) and low (beginning at 0.0) albedo value (Fig. 6(c)). For each modelled albedo, the spectral area is found in the same manner as was done for the non-simulated measurements. The modelled areas are compared against the measurement area, and the albedo value with the lower percent error is taken as the current albedo estimation (Fig. 6(d)). If agreement is not within 3% new high and low albedo values are calculated by perturbing the estimated albedo proportionally to twice that of the percent difference, and the process repeated.

**Figure 6.** Flow diagram of the albedo estimation algorithm. (a) ALI radiance profiles collected and mean radiance values found. (b) The values collected in step (a) as a function of wavelength (dotted line), regressed (solid line), and integrated (shaded region) making a metric of albedo. (c) The configurations of step (a) are forward modelled for two albedo values and processed as step (b) was. (d) Modelled areas of step (c) compared against area of step (b) to estimate the albedo. The cycle of steps (c) and (d) are repeated until agreement is found.

As an example evaluation, ALI was simulated (with instrument noise) measuring a forward modelled atmosphere of known true state surface albedo of 0.6, and inclusion of a GloSSAC (Thomason et al., 2018) aerosol extinction profile. Then ALI (without noise) is modelled observing a Rayleigh atmosphere which has the albedo adjusted following the flow diagram of Fig. 6. This exercise yielded the final effective albedo estimation of 0.654, which we consider to be a reasonable estimation given the parameters of the true state. Following this, we show the estimated albedo of each example ALI scan in Table 1. Note that Scan 2 proved insensitive to changes in albedo, which is expected from sunrise conditions, so we simply assign a value of 0.3 here. However, the low sun condition also means the retrieval is quite insensitive to the large uncertainty in albedo for this case.

# 4.2 Cloud Discrimination



A primary motivation for the polarimetry of ALI is to discriminate scattering by clouds and scattering by aerosol. This is so the contribution of cloud scattering is not attributed to atmospheric aerosol in the retrieval. Typical approaches to this problem study the vertical gradient of limb radiance profiles and how it differs over wavelength (Chen et al., 2016), but this is still prone to identifying aerosols with larger particle sizes as clouds. However, scattering by clouds will tend to reduce the DoP (Hansen, 1971; Deirmendjian, 1964), so relative changes in polarized light can be used as a metric to determine if limb-scattered signal was influenced by cloud or not.

The goal of the analysis here is to acquire a vertical profile of the DoP. While this is useful in itself to identify altitudinal regions of different scattering behaviour, for the purposes of the present work we focus only on determining the lower limit of the aerosol retrieval. That is to say, quantitatively identify a tangent altitude in which cloud scattered light becomes significant as indicated by a reduction in the atmospheric DoP. Aerosol retrieved using measurements above this tangent altitude will be free of any notable ambiguity with cloud.

The atmospheric DoP can be directly approximated from ALI measurements, which Section 4.2.1 discusses. Unfortunately, the non-ideal behaviour of the ALI polarimetric response reduces the effectiveness of this approximation. Therefore, we also present an approach to retrieve the Stokes parameters of the atmosphere in the Stokes basis of ALI in Section 4.2.2. This DoP

retrieval provides a more robust analysis of the atmospheric DoP and is the method we employ for cloud identification in Section 5.

# 4.2.1 Direct Approximation of DoP




As discussed in Section 2, ALI measures two different polarization states for each wavelength  $\lambda$  depending on if the LCR is engaged or not. Ideally, the measurement with the LCR off  $(I'_{LCRoff})$  would have ALI measure horizontally polarized light  $(|E_x|^2)$ , while the LCR on measurement  $(I'_{LCRon})$  would measure vertically polarized light  $(|E_y|^2)$ . However as Section 2.2 addresses, the Mueller matrix of ALI does not yield this ideal response and the performance is wavelength dependent. Despite this, vertical profiles of the Stokes parameters I and Q, as well as the DoP (P) can be approximated (noted with  $\tilde{I}$ ,  $\tilde{Q}$ , and  $\tilde{P}$  respectively) using the  $I'_{LCRon}$  and  $I'_{LCRoff}$  observations like those shown in Fig. 5. Equations 5 to 7 show these relations.

$$I(\lambda) = |E_x(\lambda)|^2 + |E_y(\lambda)|^2 \qquad \approx \tilde{I}(\lambda) = I'_{\text{LCRoff}}(\lambda) + I'_{\text{LCRon}}(\lambda)$$
(5)

$$Q(\lambda) = |E_x(\lambda)|^2 - |E_y(\lambda)|^2 \qquad \approx \tilde{Q}(\lambda) = I'_{LCRoff}(\lambda) - I'_{LCRon}(\lambda)$$
(6)

$$P(\lambda) = \frac{\sqrt{Q(\lambda)^2 + U(\lambda)^2 + V(\lambda)^2}}{I(\lambda)} \approx \tilde{P}(\lambda) = \frac{|\tilde{Q}(\lambda)|}{\tilde{I}(\lambda)}$$
 (7)

#### 4.2.2 Retrieval of DoP

To compensate for the polarimetric response of ALI which Equations 5 to 7 fail to do, we present an approach to retrieve the Stokes parameters of the atmosphere in the Stokes basis of ALI. Here we conceptualize the inverse problem as largely separated from the physics of radiative transfer (unlike Section 4.3) and consider each wavelength independently of the others. That is to say the measurement vector is the LCR on and off measurements at only one wavelength, and the retrieval is repeated separately for each wavelength.

An individual  $\lambda$  selected for analysis will have its measurement vector  $y(\lambda)$  constructed as

$$\boldsymbol{y}(\lambda) = \begin{bmatrix} I'_{\text{LCRon}}(\lambda)_{(0)} & \dots & I'_{\text{LCRon}}(\lambda)_{(n)} & I'_{\text{LCRoff}}(\lambda)_{(0)} & \dots & I'_{\text{LCRoff}}(\lambda)_{(n)} \end{bmatrix}^T$$
(8)

where the numbered indices n indicate detector pixels which directly correspond to tangent altitudes at the time the observation was taken. We then define the state vector as attitudinal profiles of Poincaré parameters (Bass et al., 2010) as

$$\boldsymbol{x}(\lambda) = \begin{bmatrix} I(\lambda)_{(0)} & \dots & I(\lambda)_{(n)} & P(\lambda)_{(0)} & \dots, P(\lambda)_{(n)} & \theta(\lambda)_{(0)} & \dots & \theta(\lambda)_{(n)} \end{bmatrix}^T \tag{9}$$

where  $I(\lambda)$  is the Stokes parameter I of the atmosphere at  $\lambda$ ,  $P(\lambda)$  is the degree of polarization at  $\lambda$ , and  $\theta(\lambda)$  is the orientation of the polarization ellipse major axis at  $\lambda$ . With an assumption of no circularly polarized light, the Poincaré latitude is taken as zero. Describing the Stokes parameters with respect to the Poincaré sphere provides a convenient framework for enforcing

the constraints of the system (such as  $DoP \le 1$ ) in the forward modelling. With this, the forward model of each pixel (tangent altitude) n of the retrieval can then be described as

$$I'_{\text{LCRon}}(\lambda)_{(n)} = \begin{bmatrix} 1 & 0 & 0 & 0 \end{bmatrix} \mathbf{M_{on}}(\lambda) \begin{bmatrix} I(\lambda)_{(n)} & I(\lambda)_{(n)} P(\lambda)_{(n)} \cos(2\theta(\lambda)_{(n)}) & I(\lambda)_{(n)} P(\lambda)_{(n)} \sin(2\theta(\lambda)_{(n)}) & 0 \end{bmatrix}^{T}$$

$$(10)$$

$$I'_{\text{LCRoff}}(\lambda)_{(n)} = \begin{bmatrix} 1 & 0 & 0 & 0 \end{bmatrix} \mathbf{M_{off}}(\lambda) \begin{bmatrix} I(\lambda)_{(n)} & I(\lambda)_{(n)} P(\lambda)_{(n)} \cos(2\theta(\lambda)_{(n)}) & I(\lambda)_{(n)} P(\lambda)_{(n)} \sin(2\theta(\lambda)_{(n)}) & 0 \end{bmatrix}^{T}$$

$$(11)$$

where  $\mathbf{M_{on}}(\lambda)$  and  $\mathbf{M_{off}}(\lambda)$  are the full descriptions of the ALI Mueller matrix in the LCR on and off states for  $\lambda$ . These are obtained from the work of (Letros et al., 2024) which is directly applicable here.  $\mathbf{S_{\epsilon}}$  of this inversion is constructed identically to that described in Section 4.3, that is to say a diagonal matrix of the measurement noise.  $\mathbf{S_a}$  is also a diagonal matrix with standard divinations of  $\sigma_I = 0.005$ ,  $\sigma_P = 0.05^\circ$ , and  $\sigma_\theta = 0.1^\circ$  which were selected via prototyping the algorithm (evaluating different settings in simulation). Finally,  $\mathbf{D}$  is constructed from the diagonal of  $\mathbf{K}^T\mathbf{S_{\epsilon}}\mathbf{K}$ , where naturally  $\mathbf{K}$  is constructed from the derivatives of Equations 10 and 11.  $\mathbf{D}$  is then additionally altered to decrease the dampening of  $I(\lambda)_{(n)}$  for n corresponding to tangent altitudes below 15 km, as well as increase the dampening of  $\theta(\lambda)_{(n)}$ . The purpose of this is so the beginning iterations of the retrieval will favour attributing large changes of  $\mathbf{y}(\lambda)$  expected from clouds to a change of the total light instead of the polarization parameters (i.e. favour depolarized the radiance). On further iterations  $\gamma$  will tend to zero disabling this dampening effect.







A radiative transfer model is not directly used in this retrieval, but SASKTRAN is used as a tool for constructing a-priori state profiles at different solar geometry, as well as prototyping Stoke behaviour of atmospheric radiance with respect to different atmospheric conditions. This prototyping indicated that at fixed solar geometry, the  $\theta$  state is reasonably insensitive (

Figure 7. Summary of the atmospheric Stokes retrieval used for cloud discrimination using 1105 nm observations as an example. (a, b, c, d, e) The Stokes and Poincaré parameters of the atmospheric state, where  $x_a$  (dotted cyan) represents the a-priori state, x (solid blue) represents the final retrieved state, and  $\hat{x}$  (dashed red) represents the true state of the simulation. (f, g) The profiles of y where  $\mathbf{F}$  represents the forward modelling profiles using Equations 10 and 11. (h, i) The demonstration of the cloud identification using a stark change in DoP as indication of cloud scattering.

include  $\theta$  for application on the real measurements in Scans 1, 2, and 3 to match the measurement vectors. This may indicate left over instrument biases in the calibrated profiles that are not forward modelled correctly, or that the polarimetry of the real atmosphere is not captured as well by the constructed a-priori profiles as the prototyping indicated. In either case, we do not find this an impactful issue for the determination of cloud scattering tangent altitudes.

#### 355 4.2.3 Cloud Identification from the DoP

We examine the well retrieved DoP shown in (d) of Fig. 7 to set a lower altitude limit of the aerosol retrieval. We use a simple edge detection algorithm after first smoothing the DoP (dots of (h) in Fig. 7) using a Savgol filter (producing the solid line of (h) in Fig. 7). Next we convolve the smoothed DoP with a central difference impulse response to identify a stark change in polarized behaviour. We take the higher altitude of the full-width half-max of the peak as the indication that the scattered

**Figure 8.** DoP retrieval (similar to plot to (d) of Fig. 7) at 865 nm. The a-priori DoP (dotted cyan) shown along with the true DoP of the simulation (dashed red) and the retrieved DoP (solid blue). The approximation of the DoP ( $\tilde{P}$ ) made directly from ALI observations (Equation 5 to 7) is shown as the dashed-dot orange profile. The approximation gives a very small and incorrect profile, but the retrieval method yields a much more robust and correct result.

signal is now contaminated with the presence of cloud. This process is demonstrated in (h) and (i) of Fig. 7, which arrived at an answer of 13.8 km. It is known from the true state of this simulated exercise that the depolarizing ice layer is just slightly below this at 12 km to 13km, thus making this a satisfactory indication.

A reader may wonder why the effort to retrieve the Stokes profiles of the atmosphere is justified if only an edge in the DoP is used to identify a cloud deck altitude, since one may expect that a very similar edge is also seen in the DoP approximation provided by Equations 5 to 7. While indeed the approximate measure of  $\tilde{P}$  yields a similar answer for the example in Fig. 7, one needs to emphasise that the  $\lambda$  dependent non-ideal behaviour of the LCR affects the ability to do this. For example, Fig. 8 shows the DoP for the same exercise of Fig. 7 except now at 865 nm instead of 1105 nm, and with the inclusion of  $\tilde{P}$  shown as the orange line. Referring to Equations 1, 10 and 11 the radiance profiles measured by ALI in the LCR on and off configurations depend on the combined response of the Mueller matrix of ALI, and that of the polarized state of light being observed. As the  $\tilde{P}$  of Fig. 8 indicates, in this scenario the combined response at 865 nm comes close to looking identical between LCR on and LCR off measurements and yields a very small (and incorrect) DoP compared to that of the true state when directly approximated. However, the retrieval method is robust enough to still arrive at the true state and provide a better quantification of the DoP.

# 4.2.4 ALI DoP Limitations



While the retrieval method is more robust than the approximations of Equations 5 to 7, it is still limited. Observations of  $I'_{LCRoff}$  and  $I'_{LCRon}$  result from the combined wavelength dependent response of the non-ideal ALI Mueller matrices, and that of the polarized light in the atmosphere. The combined response can lead to similarity (at least at some wavelengths) between  $I'_{LCRoff}$ 

and  $I'_{\rm LCRon}$  measurements that construct  $y(\lambda)$  of the retrieval through Equations 8, 10 and 11. If these measurements are similar than polarization cannot be distinguished.

We demonstrate this in simulation where the geometry and solar conditions of each scan of Table 1 is used and the DoP is retrieved. At each scan, the true state of the atmosphere includes GloSSAC aerosol but unlike the exercise of Fig. 7 and Fig. 8 no ice layer is included. Otherwise, the approach is the same as already discussed. This simulation is run twice, once with the polarimetric response of ALI, and again with an "ideal ALI" behaving as a perfect vertical or horizontal polarizer for each respective LCR state. We show the results in Fig. 9.

As these results show, when ALI behaves as ideal linear polarizers the DoP retrievals of all three scans ((f, g, h) of Fig. 9) fall well within 5% of the true state DoP values. This is because there is no practical potential of LCR on and off measurements looking similar and the atmospheric DoP can be well resolved. However, the non-ideal behaviour of the LCR (particularly in the off state) causes ambiguity for Scan 1 and Scan 3 shown in (b) and (d) of Fig. 9. Here the ambiguity manifested at the shorter wavelengths of Scan 1 and gave a nearly fully polarized DoP retrieval, where as Scan 3 the ambiguity at the longer wavelengths yielded an almost completely randomly polarized atmosphere. However, in Scan 2 (shown in (c) of Fig. 9) the atmospheric Stokes parameters being measured by ALI did not produce ambiguity after transformation by the Mueller matrices, and the DoP is still resolved at all wavelengths.

#### 4.3 Aerosol Retrieval






In this section we discuss the performance of the ALI aerosol retrievals in simulation against known true state aerosol. We would again like to emphasise that in the context of the present work, the algorithmic criterion we set is to retrieve an altitude resolved unimodal log-normal aerosol population, where both the altitudinal number density and median radius are retrieved along side a scalar width. To begin, we define the state vector of the aerosol retrieval to be the vertically resolved number density N (units of cm<sup>-3</sup>), and median radius r (units of  $\mu$ m) of a unimodal log-normal aerosol profile. Unless explicitly noted otherwise, the width w of this log-normal distribution is also retrieved, but only as a single scalar value which is applied to all altitudes. In prototyping, an effort was made to retrieve a vertically resolved width profile along with the number density and median radius. However, we found that while the true state aerosol extinction was well retrieved, there is simply too much freedom in the state solution space to arrive at any viably robust solution of the state properties themselves from ALI measurements. This is an unsurprising conclusion given other similar efforts (Rieger et al., 2014; Malinina et al., 2018). Therefore, we limit our retrieval state vector x to just the properties of N, r and (scalar) w as:

$$x = \begin{bmatrix} N_{\text{low alt}} & \dots & N_{\text{high alt}} & r_{\text{low alt}} & \dots & r_{\text{high alt}} & w \end{bmatrix}^T$$
 (12)

where the lowest altitude is determined by the lowest observed tangent altitude not considered contaminated by cloud scattering as discussed in Section 4.2. As for the high altitude limit, the gondola of the example science scans was at a float altitude between 36 km and 37 km, which allows for the possibility of retrieving nearly up to these altitudes. However aerosol number density can be very small at altitudes above 30 km, and results from prototyping our retrieval algorithm showed that retrieving

**Figure 9.** Error of DoP retrievals against the combined response of the ALI Mueller matrix and the simulated atmospheric Stokes for all three scans of Table 1. (a) The ALI Mueller matrix coefficients for LCR on and off states (same Mueller response shown in Fig. 3). (b, c, d) The error in the DoP retrieval as a percent change from the known true state of the simulation using the response in (a) for Scan 1, Scan 2, and Scan 3 respectively. (e) Ideal Mueller coefficients for LCR on and off states. (f, g, h) The error in the DoP retrieval as a percent change from the known true state of the simulation using the response in (e) for Scan 1, Scan 2, and Scan 3 respectively. Results at different wavelengths shown by the coloured lines in (b, c, d, f, g, h) with the shaded regions indicate one sigma of uncertainty in the retrieved DoP profile.

aerosol where the density approaches zero yields very large uncertainties in the median radius. Therefore, for the context of the present work we generally select 30 km as the ceiling of the retrieved state vector. Due to this, we also limit the ceiling on the radiance profile which construct *y* at this altitude as well.

Table 2. Diagonal a-priori covariance values





| Altitude m              | 5500.0                | 10000.0               | 22500.0               | 30000               |
|-------------------------|-----------------------|-----------------------|-----------------------|---------------------|
| Number Density Variance | $200 \ {\rm cm^{-6}}$ | $100 \ {\rm cm^{-6}}$ | $10~\mathrm{cm^{-6}}$ | $0.2~{\rm cm^{-6}}$ |

A-priori variance used to construct the diagonal elements of  $\mathbf{S_a}$  corresponding to number density. Values are specified against SASKTRAN altitude, and are interpolated onto the forward modelling grid.

Of note, in our forward modelling SASKTRAN calculates the radiative transfer for altitudes between  $0.5 \, \mathrm{km}$  to  $45 \, \mathrm{km}$  at  $500 \, \mathrm{orders}$  of scatter. The aerosol outside of the actively retrieved altitudes is scaled for altitudes below the lower altitude limit, and fixed to be zero in number density above the retrieval ceiling. Furthermore, in our retrieval the vertical resolution of the state vector effectively matches the resolution of the altitude grid in SASKTRAN. We set the discrete altitude grid of SASKTRAN to be  $0.5 \, \mathrm{km}$  to  $45 \, \mathrm{km}$  in steps of  $0.6 \, \mathrm{km}$ , where the  $0.6 \, \mathrm{km}$  resolution was determined as the finest resolution the averaging kernel is capable of producing given the content of the y we employ. Increasing the resolution of the state grid to be finer does not improve the resolution indicated by the averaging kernel.

We then construct  $S_a$  as a diagonal matrix with selected variances. For the number density state we select the a-priori variance such that the retrieval is stabilized in the high-altitude region where the aerosol number density is small, as well as the SNR of y being relatively smaller (providing poorer conditions for the inversion to work at these altitudes). Table 2 shows the variances we use along the diagonal of  $S_a$  corresponding to the number density state property and its altitude. The a-priori variance of the median radius is made uniform with respect to altitude and selected to be 0.01  $\mu$ m<sup>2</sup>. The scalar width has this variance set to 0.0001.

The measurement vectors y are simply stacked vertical radiance profiles, with  $S_{\epsilon}$  constructed as a diagonal matrix containing the variances associated with each element of y. No processing is done to the measurements of y for the sake of the inversion beyond truncating the tangent altitudes to only the region between the low-altitude and high-altitude cut-off, and high-altitude normalization. High-altitude normalization is done primarily to compensate for albedo effects which are not well encapsulated by forward model using the values determined in Table 1. The normalization itself is done by dividing by the mean signal level of each radiance profile between the tangent altitudes of 30 km and 33 km. The error associated with each point of the profile is then scaled to conserve the relative SNR at each tangent altitude of the measurement.

Additionally, while a full science scan of ALI consists of the 10 wavelengths mentioned in Section 3 in both LCR on and off states we have chosen not to use them all here. In the context of the present work we focus on using the measurements provided by the on wavelengths of  $750~\mathrm{nm}$ ,  $1025~\mathrm{nm}$ , and  $1230~\mathrm{nm}$  of the LCR on state. The reason for this restriction is that including the other wavelengths and LCR states (or the Stokes parameters of Section 4.2) within y did very little to increase the information content of the retrieval in prototyping, at least within the scope of our approach of retrieving a unimodal lognormal aerosol distribution. Furthermore, as mentioned before the other wavelengths can introduce further complexity as the wings of the ATOF bandpass have sensitivity to trace gas absorption which needs further and careful analysis to handle.

The dampening matrix  $\mathbf{D}$  is constructed similar to the dampening matrix discussed in the context of Section 4.2, that is we construct  $\mathbf{D}$  in state space as a diagonal matrix populated with the diagonal values of  $\mathbf{K}^T \mathbf{S}_{\epsilon} \mathbf{K}$ . However, unlike Section 4.2 the dampening matrix is not further configured. This matrix is paired with a starting  $\gamma$  of 1.0 which is adjusted according to the minimization of the cost function.

# 4.3.1 Aerosol Retrieval Simulations




We now present a summary of retrieval results obtained in pure simulation where the true state aerosol is known. In all of these simulations measurement noise is applied to the simulated observations and we construct the true state aerosol profile from GloSSAC extinctions for realistic aerosol scattering. For this true state aerosol, we customize r and w, and then adjust N such that the GloSSAC extinction is conserved at 525 nm. The main exception to this conservation of GloSSAC extinction is we force the true state number density above 30 km to be zero. We do this because of our use of high-altitude normalization, which makes this assumption implicit.

Using the geometry of Scan 3, we first show a simplified case of our standard retrieval approach in which only the vertical profile of N and r is retrieved. In this specific exercise w is fixed at what normally is our a-priori value of 1.6 for both the true state and the retrieval forward modelling. We also choose for our a-priori and initial state a uniform r of 0.08  $\mu$ m, and an a-priori N profile which is shown alongside our exercises. However, we note that this a-priori N profile is a very simplified expectation of aerosol number density. It is not constructed from any specific knowledge of the aerosol to be retrieved (no a-priori refinement from other instrumentation or sources like GloSSAC). These a-priori values are used in all exercises (real and simulated) for the remainder of the present work except Section 5.1. This retrieval is shown in Fig. 10 where the GloSSAC aerosol extinction is obtained by retrieving N and r profiles, both of which are well representative of the true state parameters.

Now we demonstrate the efficacy when the retrieval of Fig. 10 is repeated, but with the addition of the scalar w re-included in x as our nominal approach uses. The true state width is made scalar at 1.5. The state results are shown in Fig. 11. This simulation represents the *viable* limits of our approach. Note that while the true state extinction is well retrieved and y is well agreed, the scalar width was unable to obtain the correct value despite the true state width also being scalar. Furthermore, the shape of both the N and r states is well represented and only separated from the true state by the biases caused by the incorrect retrieval of w. Essentially the retrieval found an aerosol particle size and number density which reproduces the ALI observations while not being faithful of the true state. Despite this however, the retrieved w is still an improvement over our a-priori w, and because of this we consider this a better retrieval approach over assuming a fixed width.

Our final two simulations we present show the behaviour of the algorithm in the presence of more complex aerosol distributions - which we present to contextualize some results in Section 5. In this exercise the ground truth aerosol is bimodal with N, r, and w all varying in altitude. We then apply our retrieval algorithm assuming a unimodal distribution with a scalar w in two cases. The first case uses the observational and solar geometry of Scan 3 (as the retrievals of both Fig. 10 and Fig. 11 used), where the polarization of limb-scattered sunlight is expected be horizontally dominated. The second case uses the solar geometry of Scan 1, where the limb viewing geometry is expected to yield a *relatively* equal balance between horizontal and vertically polarized light.

**Figure 10.** Simulated ALI unimodal log-normal aerosol retrieval under a simplified approach which fixes the aerosol distribution width at 1.6. (a) Profile of aerosol number density. (b) Profile of aerosol log-normal median radius. (c) Profile of aerosol log-normal width. (d) Profile of aerosol extinction. In (a,b,c,d) the red lines represent the true state of the simulation, and the cyan dots show the a-priori and initial state of the retrieval. The dashed blue lines show the state being modelled in SASKTRAN outside of our active retrieval altitudes, while the solid blue line shows the retrieval itself. Note that the blue shaded region represents 1 standard deviation of uncertainty for retrieved each state. (e) Measurement vector of 750 nm. (f) Measurement vector of 1025 nm. (g) Measurement vector of 1230 nm. In (e, f, g) all profiles are from LCR on observations. The cyan dotted line is the forward modelled vector given the a-priori state, the dashed blue line is the forward modelled vector of the final retrieved state, and the orange line is the actual vector made from ALI observation (simulated from the true state atmosphere). Error bars of the measurement are shown, but too small to be easily visible. This relatively small error of the measurement vectors primarily results from the column binning of the ALI images to produce the radiance profiles. (h) The percentage difference of the retrieved aerosol extinction from the known true state of the simulation.

The results of the first case using Scan 3 geometry is shown in Fig. 12. It is rather clear that the retrieval algorithm we present fails to arrive at a representative atmospheric state which can well reproduce all of the ALI observations of the more complex aerosol. In particular, the retrieved extinction underestimates the total true state extinction of the bimodal distribution shown as the red line in Fig. 12. For clarity, the inversion itself worked as intended but was unable to produce a more optimized

**Figure 11.** The unimodal log-normal aerosol retrieval of Fig. 10 but now retrieving the width (our nominal approach). Descriptions of (a, b, c, d, e, f, g, h) same as Fig. 10. Important here is the measurement vectors and aerosol extinction profile agree well, but the retrieved N, r, and w are biased from the true state. The retrieval has determined a different aerosol population which still reproduces the ALI observations.

x than what is shown under these conditions. However, when we repeat the retrieval of the exact same bimodal distribution - changing only the geometry to that of Scan 1 - the results shown in Fig. 13 are obtained. While there are still inaccuracies of this retrieval, particularly in the reproduction of y using the retrieved representative unimodal distribution, the overall performance significantly improved over using the geometry of Scan 3. Of particular note, the retrieved extinction very well represents the true state of the bimodal distribution.



We wish to emphasize that the limb measurements of ALI are polarized, and this polarized content contains useful information about the aerosol phase scattering matrices - which is of course influenced by the particle sizes. For example, the state properties of the GloSSAC profile used in the DoP exercises Fig. 7 and Fig. 8 is the same as what is shown in Fig. 10. The feature seen in the DoP around 18 km of these two figures corresponds to the feature of the median radius in the true state aerosol. Regardless, the more the limb-scattered radiance is polarized the more pronounced the requirement of accurately modelling the aerosol scattering matrices is. This may yield potential to retrieve more complex aerosol distributions with more complex retrieval

**Figure 12.** Unimodal aerosol retrieval algorithm performance under observation of a bimodal distribution using the geometry of Scan 3. The solar scattering angles of Scan 3 are expected to give horizontally dominated limb-scattered radiance. Descriptions of (a, b, c, d, e, f, g, h) same as Fig. 10, except (a, b, c, d) now show the true state of the bimodal distribution in orange and green. The red line in (d) shows the total aerosol extinction of the bimodal distribution, and it is used as the true state for comparison in (h).

approaches. However this is a point of on-going research and not within current scope. The relevant conclusion to be made from the results of Fig. 12 and Fig. 13 is that through prototyping our algorithm, we expect a unimodal distribution to be more representative of a complex aerosol population the less polarized the observations are.

#### 5 Aerosol Retrievals of the Flight Campaign


With efficacy of our approach evaluated in Section 4, we now show its application to the exemplar ALI observations summarized in Table 1 made during the last high-altitude balloon flight. We begin with the cloud discrimination by retrieval of the DoP profiles, shown for all three scans in Fig. 14. From this analysis, cloud contamination begins at a tangent altitude of 10.8 km in Scan 1 and 10.2 km in Scan 2. As expected from the example image of Scan 3 in Fig. 5, it did not produce a change in the

**Figure 13.** Retrieval of Fig. 12 repeated using the geometry of Scan 1. Descriptions of (a, b, c, d, e, f, g, h) same as Fig. 12. The solar geometry of Scan 1 is expected to yield a generally neutral balance between vertically and horizontally polarized limb-scattered light. With respect to Fig. 12, the change in geometry leading to a less polarized observation allowed the unimodal assumption to perform better.

DoP profile that would indicate the presence of significant cloud. For the purposes of this scan we simply select a lower limit of 10 km only for consistency with Scan 1 and 2.

Furthermore, as mentioned before we note that Scan 3 is the noticeably more polarized than Scan 1 or 2, and Scan 1 is the least polarized. With respect to simulations, we find almost all behaviour regarding Fig. 14 to be expected including: the relative balance between horizontal and vertical polarizations for all three scans, the spectral regions expected to fail given the polarimetric response of ALI in Scan 1 and Scan 3, and the relative magnitude of the DoP for Scan 1 and Scan 2. However, a notable exception to our expectations is the magnitude of the retrieved DoP in Scan 3 of Fig. 14. The simulation of this geometry produced a true state DoP approximately ranging between 0.3 - 0.4 (with the GloSSAC aerosol loading used in Fig. 10), but the corresponding DoP in Fig. 14 is significantly larger. We find no indication that the discrepancy is erroneous and consider that the increased DoP is a measured feature of the atmosphere, but the specific cause is still under investigation.



**Figure 14.** DoP profiles of all three scans found following the technique shown in Section 4.2. (a) Results of Scan 1. (b) Results of Scan 2. (c) Results of Scan 3 (no significant cloud). Dashed black line indicates the tangent altitude of cloud contamination. Coloured lines show analysis at different wavelengths. Wavelengths not shown were too similar between LCR on and LCR off states of ALI to distinguish atmospheric polarization.

As we apply our retrieval approach to each scan, we compare our results to the extinction of three other instruments: SAGE III/ISS, OMPS-LP, and OSIRIS. In this comparison, we convert our unimodal log-normal state parameters to an extinction at 750 nm for relevant comparison. However, to first establish the initial footing of this comparison we begin by simplifying the retrieval approach we have discussed in Section 4.3, and apply only our 750 nm measurements to retrievals with fixed r of 0.08  $\mu$ m and fixed w of 1.6. This is a similar approach to the standard retrieval approach of OSIRIS and OMPS-LP (Rieger et al., 2019; Taha et al., 2021). In this simpler retrieval only N is retrieved in x, with only the 750 nm radiance constructing y. Since the observations of ALI are polarized, we attempt to compensate in this simplified retrieval by constructing y using a 750 nm Stokes parameter I profile built with the approximation of Equation 5. We use  $\tilde{I}$  since the retrieval of I used in the cloud discrimination is not available for Scan 1 at 750 nm. Figure 15 shows the results of this exercise. Retrieving extinction (by retrieving N using fixed aerosol size) with only 750 nm yields encouraging results for all three scans, but we note that with respect to the other instruments our retrievals tend to over estimate the extinction in the lower altitudes.




In the retrievals of Fig. 15, the forward modelling of ALI observations is still all polarized appropriate to the ALI flight observations (i.e. the construction of  $\tilde{I}$  in Equation 5). This does still present a polarized aspect to the ALI retrievals (within the forward modelling and measurement vector construction) of this exercise which hampers a completely level methodology to the comparison. Regardless of this however, we note that an overestimation of the aerosol extinction may be a result of the limitations imposed on a ballooning platform as opposed to a satellite. The balloon float altitude, varying between 36 and 37 km, limits the hight of limb observations to about this altitude. This altitude limit makes retrieving albedo difficult due to ambiguous signal (hence the approach of Section 4.1 removing it from the state vector) and emphasizes the need to high-altitude normalize the measurement vectors to compensate. Normalization of the measurement vectors implicitly invokes

Figure 15. Extinction retrievals from each ALI scan using only  $\tilde{I}(750 \text{ nm})$  built with Equation 5 compared against extinction profiles of SAGE III/ISS, OMPS-LP, and OSIRIS. (a) Extinction retrieval of Scan 1. (b) Extinction retrieval of Scan 2. (c) Extinction retrieval of Scan 3. (d) The agreement of the Scan 1 extinction shown in (a) as a percent difference of the ALI extinction profile with respect to SAGE III/ISS, OMPS-LP, and OSIRIS extinction. Coloured lines of (d) correspond to legend in (a). (e, f) Depict the same as (d) except (e) corresponds to Scan 2 of (b) and (f) to Scan 3 of (c). (g) ALI measurement vector corresponding to extinction retrieval of Scan 1. (h) ALI measurement vector corresponding to extinction retrieval of Scan 3. In (d,e,f) the cyan dotted line is the forward modelling of the measurement vector given the a-priori state, the dashed blue line is the forward modelled measurement vector of the final retrieved state, and the orange line is the actual measurement vector made from ALI observation.

the assumption that the aerosol in the forward modelling at these altitudes is correct to the atmosphere, else a bias will be introduced to the retrieval. As already discussed, we assume aerosol as zero above our 30 km normalization in this work. While we do not consider this unrealistic we also understand it may not be strictly correct. These effects are in addition to other potential culprits such as uncertainties in the attitude solution of the gondola, instrumentation biases not correctly removed during calibrations, or other aspects of the atmosphere which have not been properly accounted for in the forward modelling. Therefore, given the results of Fig. 15 and the context of this discussion, we acknowledge that the ALI aerosol extinction presented in this work tends to bias high compared to SAGE III/ISS, OMPS-LP, and OSIRIS. However, we do not feel this bias invalidates the underlying demonstration of the ALI methodology.







We now show our algorithm which retrieves N, r, and a scalar w using the LCR on measurements of 750 nm, 1025 nm, and 1230 nm of ALI applied to all three scans of Table 1. We present first the retrieval of our nominal scan, Scan 1 shown in Fig. 16. Here all three measurement vectors produced by the retrieved aerosol state represent the ALI observation of the flight very well. Furthermore, we find the comparison of the retrieved aerosol extinction of ALI and the extinction profiles of all three comparison instruments very encouraging. The median absolute percent difference,  $\text{med}(|\Delta\%|)$ , between ALI and each of the comparators is between 21% and 29% (see Fig. 16(h)). We also note that the lower altitude bias of ALI with respect to the other instruments seen in Fig. 15 is brought into further agreement. While the w of this retrieval did not adjust significantly from the a-priori value of 1.6, of interest is the profile of r which indicates a layer of larger particles at approximately 22.5 km.

The retrievals of Scan 2 and Scan 3, shown in Fig. 17 and Fig. 18 respectively, maintain larger disagreement with SAGE III/ISS, OMPS-LP, and OSIRIS than Scan 1. The metric of  $med(|\Delta\%|)$  is between 35% and 47% for Scan 2 and 31% and 44% for Scan 3. However, we also observe increased disagreement between ALI radiance in these two scans with respect to the forward modelled observations produced by the retrieved aerosol state. In particular, the measurement vector of 750 nm in Scan 2 has significant difference with the ALI measurement. Here we suspect the unimodal assumption of the retrieved aerosol state is less able to perform under increasingly polarized conditions as discussed in the simulated exercise surrounding Fig. 12 and Fig. 13. Furthermore, all three measurement vectors of Scan 3 show inconsistencies similar to that observed in the simulated exercise of Fig. 12. Similar to the Scan 2 retrieval, we again see increased disagreement with respect to the retrieval of Scan 1, and suspect the same root cause.

We again recognize the possibility that disagreements shown in Fig. 16, and more so in Fig. 17 and Fig. 18 may be related to ALI bias discussion around Fig. 15. However, we find no significant reason to invalidate the bimodal influence as discussed in the exercise of Section 4.3. In pure simulation where observational geometry and forward modelling are identical between simulated observations and the inversion process, we demonstrated the creation of a similar retrieval disagreement in our approach when it is applied to more complex aerosol distributions than the retrieval assumes. This disagreement is mitigated as only the geometry is changed from Scan 3 to Scan 1, where the limb-scattering conditions produce a less polarized atmosphere. We consider that the relatively good performance of Scan 1 in Fig. 16 with respect to Scan 3 in Fig. 18 is an indication that the effect discussed in Section 4.3 related to this is manifesting.

With that said, we can also highlight positive aspects of the Scan 2 and Scan 3 retrievals. Scan 1 and Scan 3 are both looking south, so they should be observing very similar aerosol. We see this represented in the similar shapes of the states between

**Figure 16.** Aerosol retrieval of Scan 1 (nominal conditions). (a) Profile of retrieved aerosol number density. (b) Profile of retrieved median radius. (c) Retrieval of scalar width. (d) Retrieved aerosol extinction profile of ALI shown with SAGE III/ISS, OMPS-LP, and OSIRIS. (e, f, g) Retrieval measurement vectors as described in Fig. 10, noting that the orange is now made from ALI flight observation. (h) Shows the agreement of (d) as a percent difference of the ALI extinction profile with respect to SAGE III/ISS, OMPS-LP, and OSIRIS extinction. Coloured lines of (h) correspond to legend in (d).

these two scans - in particular the r profiles. However, we also note that in Scan 3 r at lower altitudes gets significantly smaller than in Scan 1, while N increases significantly. We speculate that this is another manifestation of the retrieval trying to optimize a unimodal distribution to match the polarized y produced by a more complicated aerosol population. In contrast, the retrieved state of Scan 2, which is looking north, is yielding a distinctly different radius profile. This indicates that even under the limitations of our approach, the retrievals are still sensitive to aerosol particle size information. Additionally, we highlight that the overestimation of aerosol extinction with respect to SAGE III/ISS, OMPS-LP, and OSIRIS seen in Scan 2 and Scan 3 is similar to what was seen in the more straightforward 750 nm extinction retrievals shown in Fig. 15. Except unlike the simplified retrievals there is now the indication of a biased state given the y disagreement with our approach.


**Figure 17.** Aerosol retrieval of Scan 2 (vertically dominate polarization). Descriptions of (a, b, c, d, e, f, g, h) same as Fig. 16. Compared with the retrieval done for Scan 1, there is increased disagreement between the ALI extinction profile in (d, h), and that of SAGE III/ISS, OMPS-LP, and OSIRIS. However, (e, f, g) also show increased disagreement between the forward modelled measurement vectors of the retrieved state and the flight observations of ALI.

#### 5.1 Retrieved Aerosol State Comparison Between ALI and SAGE III/ISS

Since the SAGE III/ISS retrieval includes aerosol size properties as well as the extinction (Knepp et al., 2024), we extended the comparison between the ALI retrieval results and SAGE III/ISS coincident profiles to the aerosol size properties for all three ALI scans. We show this under two scenarios: the first is using the direct results shown in Fig 16, Fig 17, and Fig 18 - which are retrievals using the a-priori median radius  $r_a = 0.08 \, \mu m$  and scalar width  $w_a = 1.6$  for the unimodal log-normal distribution as noted in Section 4.3.1. The second is repeating the retrievals of each ALI scan but now fixing  $w_a$  to what SAGE III/ISS indicates, and adjusting the  $r_a$ . In this latter scenario,  $w_a = 1.49$  and is not retrieved. This width is a mean value taken from the SAGE III/ISS data. The  $r_a$  is then changed to 0.093  $\mu m$  so as to maintain the same a-priori effective radius of the log-normal

Figure 18. Aerosol retrieval of Scan 3 (horizontally dominate polarization). Descriptions of (a, b, c, d, e, f, g, h) same as Fig. 16.



distribution between the two scenarios. For clarity, although the width is now fixed in this new scenario both number density N and median radius r are still retrieved as normal.

The retrieved ALI aerosol extinction of all three scans in this second (adjusted a-priori) scenario was near identical to what is shown in Fig 16, Fig 17, and Fig 18 respectively, along with near identical performance of the measurement vectors. We do not show the extinction and measurement vectors of the adjusted a-priori retrievals for brevity, but the near identical performance is expected as shown in the simulated exercise between Fig. 10 and Fig. 11 where only the underlying aerosol state properties differ noticeably. The aerosol state properties of SAGE III/ISS coincident measurements, along with the retrieved ALI aerosol state properties for both a-priori scenarios, are shown in Fig. 19.

In all three scans, and both a-priori approaches of the ALI retrieval, we note that the ALI aerosol overestimates the number density with respect to SAGE III/ISS while (generally) underestimating the size. Incorporating the indicated SAGE III/ISS width into the ALI retrieval a-priori reduced the disagreement but did not eliminate it. These results (in addition to the simulations of Fig. 10 and Fig. 11) make clear an influence of a-priori selection on the relative magnitudes the ALI aerosol state properties (N, r, and w) have with respect to each other in representing the same aerosol extinction. This is also effectively

**Figure 19.** Aerosol state properties of each ALI scan as found following two a-priori approaches of the ALI retrieval, compared to the aerosol state reported by SAGE III/ISS for each respective coincident profile. (a, b, c, d, e) Show the aerosol properties of Scan 1. (f, g, h, i, j) Show the aerosol properties of Scan 2. (k, l, m, n, o) Show the aerosol properties of Scan 3. (a, f, k) Show retrieved aerosol number density. (b, g, l) Show retrieved aerosol median radius r. (c, h, m) Show aerosol log-normal width w. (d, i, n) Show the retrieved aerosol effective radius. (e, j, o) Show the effective radius agreement between ALI and SAGE III/ISS as a percent difference of the ALI results from SAGE III. The coloured lines show SAGE III/ISS results as green, the ALI results of Fig 16, Fig 17, and Fig 18 in blue, and the ALI results with an adjusted a-priori in orange.

observed in the simulation of Fig. 11, but as that simulation also indicated the (relative) vertical shapes of the profile properties may be correct even if the absolute magnitude is not.



Although our choice of a-priori is in line with OMPS-LP (Rozanov et al., 2024), ultimately the a-priori used in this work was arbitrary. Investigating the influence of a-priori selection (under unimodal and bimodal cases), and strategies to pick more appropriate a-priori properties is a point of on-going work. Regardless, we find encouraging indication of size agreement between ALI and SAGE III/ISS with the metric of effective radius. The  $med(|\Delta\%|)$  of the effective radius between ALI and SAGE III/ISS for Scan 1 is 50% reduced to 23% with  $w_a=1.49$ , 52% reduced to 34% in Scan 2, and 58% reduced to 33%

in Scan 3. In particular, the retrieved effective radius of ALI in Scan 1 using the  $w_a = 1.49$  agrees to SAGE III/ISS within uncertainty for most altitudes as shown in Fig. 19(e).

# 6 Conclusions




We presented the atmospheric aerosol profiling capabilities of ALI which comprises two central aspects: retrieval of the atmospheric DoP to determine influence of cloud-scattered radiance, and a unimodal log-normal aerosol retrieval algorithm which employs the polarized radiance profiles ALI observes. The efficacy of both aspects was demonstrated in pure simulation with known true states. We find that the atmospheric DoP can be well retrieved, provided the combined response of the atmospheric Stokes parameters and the ALI Mueller matrix of LCR on and off states gives enough information to distinguish linear polarization. In this work we apply the DoP information to determine quantitatively a lower altitude limit of the aerosol retrieval which avoids cloud contamination as indicated by depolarization.

The unimodal log-normal aerosol retrieval itself retrieves aerosol number density, median radius, and a scalar width (applied to all altitudes of the distribution). While we find limitations to this technique which we discuss in Section 4.3, we see good performance when the assumption of a unimodal aerosol distribution is true. When a bimodal aerosol distribution is present under a unimodal assumption, we find the performance of our retrieval algorithm worsens as the polarization of the atmosphere increases. However, an indication of this failure is the forward modelled measurement vectors of the unimodal state do not reproduce the simulated measurement vectors of the true bimodal state. This itself is a limited indication of aerosol size properties. We speculate that the polarized radiance profiles of ALI contain useful information relating to the phase scattering matrices of the aerosol population, and as the atmosphere becomes more polarized the importance of accurately forward modelling the phase matrices (i.e. particle size) increases, and the more the unimodal assumption breaks down. This indicates further potential to retrieve more complex aerosol distributions using ALI observations with a more sophisticated retrieval algorithm, but this is a point of our on-going research.

We conclude by applying our algorithm to retrieve the atmospheric aerosol observed by ALI during a high-altitude balloon flight in August of 2022. The results of three distinct sets (Scan 1, Scan 2, and Scan 3) of ALI observations are compared to the nearest coincident aerosol extinction profiles of SAGE III/ISS, OMPS-LP, and OSIRIS. We note that the retrieved ALI extinctions are biased high with respect to the other three instruments, possibly as a result of limitations imposed by the ballooning platform as discussed in Section 5. However, we find in this study that our nominal observations of Scan 1 produced a retrieved aerosol extinction in decent agreement to all three other instruments (med( $|\Delta\%|$ ) between 21% and 29%) with this consideration. Contrasting this, Scan 2 and Scan 3 showed an increased overestimation of aerosol extinction with respect to SAGE III/ISS, OMPS-LP, and OSIRIS (med( $|\Delta\%|$ ) is between 35% and 47% for Scan 2 and 31% and 44% for Scan 3). However, the increased disagreement of these two scans can potentially be explained by the effect of using a unimodal distribution to represent more complex aerosol under polarized limb measurement conditions. We speculate that the same underlying retrieval behaviour seen between the simulated exercises of Fig. 12 and Fig. 13 may be occurring in the real

retrievals as well. Supporting this statement is the relative improvement in the quality Scan 1 exhibits which is also replicated in simulation under this case.

Furthermore, the ALI polarized retrievals of Scan 1, Scan 2, and Scan 3 yielded particle size information about the aerosol population in addition to the extinction. When comparing ALI aerosol number density, median radius, and width to that reported by SAGE III/ISS, we find that the ALI results overestimate number density and underestimate the aerosol size relative to SAGE III/ISS. However, adapting the log-normal width indicated by SAGE III/ISS into the a-priori of the ALI retrieval reduces the disagreement. In the nominal case of Scan 1, this adoption yielded a retrieved aerosol effective radius which agrees to SAGE III/ISS within uncertainty for the majority of altitudes.

Data availability. Letros, D.: Aerosol Limb Imager Timmins 2022 [Data set]. Zenodo, https://doi.org/10.5281/zenodo.15707122, 2025.


630

635

Author contribution. AB and DD are the heads of ALI project. DL, LG, and PL designed and built this ALI instrument. NL and LR gave software contributions. DL developed and performed data analysis. DL wrote manuscript draft. AB and DD reviewed and edited the manuscript.

645 Competing interests. The authors declare that they have no conflict of interest.

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
