# Peer review of "The Aerosol Limb Imager: Multi-spectral Polarimetric Observations of Stratospheric Aerosol"

_EGUsphere, 2025_

## Referee Comment (RC1)

The authors have put together a very impressive paper to provide a brief description of their instrument (the Aerosol Limb Imager, aka ALI) and its aerosol retrieval algorithm. Herein, the authors provide a thorough presentation of their algorithm and discuss some of its limitations in a comprehensive manner. The ALI instrument takes advantage of limb scatter measurements at multiple wavelengths for 2 polarization states. This information content enables the authors to retrieve aerosol radius and number density estimates as well as a rather rudimentary estimate of the distribution width, which is assumed (forced?) to be constant throughout the profile.

I must reiterate that overall I am pleased with this paper. It is well written (up until section 4.3 where typos became more plentiful as did missing articles), logically organized, easily read/followed, and generally well supported with ample use of figures. I believe this paper will make a important contribution to the scientific literature, that it fits well within the scope of AMT, and should be published pending some revisions.

1. **General comment:** The authors used SAGE III/ISS, OMPS, and OSIRIS data to evaluate their derived extinctions, but nowhere do they tell the reader what profiles were used (e.g., profile or event number), do not tell the reader where these profiles were collected (e.g., lat/lon of the satellite observations), and they fail to inform the reader of when these profiles were collected. All of this information is necessary for interpreting these intercomparisons (some of this may explain the differences they observed between the satellite instruments and ALI). This information should be added before publication.

2. **General:** The authors failed to inform the reader of where ALI was sampling. The authors stated in section 3 that "ALI was situated on the balloon gondola and orientated such that when the gondola is flat and level, the highest lines of sight (top pixels of the ALI detector) would be horizontal *and with tangent locations on the instrument itself*" (emphasis added). The question is: where is the bulk of the scattering coming from? It's certainly not the "tangent point" (i.e., at the instrument itself). Maybe the distance is negligible, but the reader probably does not know that. This information should be included before publication.

3. **General:** The authors miss an opportunity to compare their radius and number density estimates with those released by the SAGE team. If there were coincident OPC data available I would recommend they use that, but since the SAGE data is all that's available I must strongly recommend the authors include that data in their analysis before publication.

4. **General/Figures:** The figures could benefit from being larger, which would enlarge the fontsize of the axes and make them easier to read. Please consider making this change.

5. **Page 6, line 122:** The authors claim that the error is "...on the order of the square root of the DN values..." Maybe I am being too precise, but the FWHM in Fig. looks to be $\approx 40$, which corresponds to a DN of 1600 (i.e., $40^2$) whereas panels (a) and (b) indicate DN on the order of 8000. Would the authors please clarify their meaning here?

6. **Fig. 4:** What is the black dot in panel (a)? That corresponds to the large white dot in panel (b), so I assume it is a bad pixel, but all other bad pixels are white. Would the authors please clarify?

7. **Page 7, lin 143:** The authors refer to ALI's viewing geometry "...when the gondola is flat and level..." I have two questions regarding this:

(a) How often is the gondola level and flat? I assume there is some stabilization utilized, but it is never explicitly mentioned.

(b) Is this orientation monitored to allow correction? Was a correction applied? How does this variation impact the view geometry and the results?

8. **Fig. 12:** I wonder if the authors sell themselves short on this figure. You cannot solve for bimodal distributions, but you may be able to reasonably infer the effective radius, which would resolve some of the "bimodal issues" the authors allude to throughout sections 5/6 as well as provide a more robust number for use in models. That said, the constant width value may skew $r_e$. I suggest the authors include another panel in this figure to evaluate $r_e$.

9. **Page 23, line 460:** The authors state "In this simpler retrieval only $N$ is adjusted in $\mathbf{x}$ such that we arrive at an aerosol extinction directly retrieved at 750 nm." As written, it sounds like the authors are iterating to match an extinction, when I think they are iterating to match the radiance (like OMPS and OSIRIS). Would you please clarify?

10. **Page 24, line 465:** The authors state "Retrieving extinction at only 750 nm yields *respectable*..." (emphasis added). "Respectable" is ambiguous, please quantify and clarify.

11. **Page 24, line 469:** The authors state "...shows a fairly ideal retrieval." What is meant by "fairly ideal"? I have no idea what is meant. Can the authors be quantitative or provide a metric for the reader to gauge "idealness"?

12. **Page 24: line 470:** The authors state "...the retrieved extinction well represents the extinction profile of all three comparison instruments." Again, "well represents" is ambiguous. The figures in reference (15, 16, 17) are plotted on log scales, which makes quantitative evaluation challenging. Readers would greatly benefit from a percent difference plot (or a ratio plot where the 3 comparison instruments are divided by ALI's extinction). This would convey a wealth of information to the reader. Would the authors please include these plots?

13. **Fig. 18:** I think the authors are overly optimistic in the interpretation of panels (a) and (b). There is no way the number density is so large (it's at least an order of magnitude too high, even after a major eruption) and the sensitivity of their instrument (at the designated wavelengths) to particles with radius of 0.04 $\mu$m is questionable. I would suggest that if the authors want readers to take panel (b) seriously then they should provide more support (maybe show scattering intensity at the current scattering angle for the wavelengths in question as a function of particle size?).

14. **Page 29, line 522:** Regarding the sentence starting with "However, the disagreement of these two...": The authors put all the blame on a bimodal distribution, which may or may not be the case. However, I don't think this is supported and I am unsure of what the authors are trying to communicate within the last 2 sentences of this paragraph. Are they suggesting that the instrument is seeing different atmospheres in the various scans and scan 2 observed aerosol with a bimodal distribution? Are they suggesting that the instrument is seeing the same atmosphere, but the profiles are different because a bimodal distribution has more impact at some scattering angles than others? I don't know. Would the authors please clarify?

---

## Referee Comment (RC2)

**General comments on Letros et al. [2025]**:

This is a well-executed paper that presents analysis of data taken by the ALI instrument, which records polarized radiance spectra. The value of these measurements is demonstrated through both sensitivity studies and analysis of selected ALI data. A particularly useful capability provided by radiance information is the ability to distinguish between clouds and aerosols along the line of sight.

I have few criticisms of the work presented, and particularly appreciate the degree of testing presented for the algorithm approach taken (which is clearly designed to test its limits, rather than primarily to show its performance in the best possible light). Error analysis is thoughtful and thorough. Some structural suggestions and clarification questions follow, but I have a positive opinion of the paper, and would be glad to see it published.

**Detailed comments:**

*Abstract, line 4:* "Tangent altitudes which have signal contaminated by clouds" should be defined more clearly. Throughout the paper, similar wording is used, and the authors' interpretation of this language appears to be consistent: This language is meant to describe tangent altitudes for which a cloud appears *along the line of sight*. But that's not the only possible interpretation: For example, limb scattering observations can (and frequently are) "contaminated" by upwelling radiation from a broad area below the line of sight, for which the locations and properties of the clouds are not well known. That is a distinct problem from interpreting measurements with clouds along the line of sight, and it isn't the focus of this paper, which the abstract should make clearer.

*Abstract, lines 14-15:* The abstract ends by declaring "good agreement" between the aerosol extinction coefficients derived here and coincident data (from SAGE III, OMPS and OSIRIS). It would be more useful to make this statement quantitative, or at least indicate what criteria were used to conclude that the agreement is "good."

The statement about particle size information would also benefit from clarification: A short summary of the particle size distribution assumptions made and the relative quality of the various retrieved properties would help a reader who is reading the abstract before deciding whether to proceed further.

*Line 47:* "purposes of the SPS is" – should be "are"

*Figure 2:* The spectral response functions shown in parts (a) and (b) appear to be fairly Gaussian – it would be useful to include an estimate of the FWHM (or some other indication of their width). The caption mentions that the FWHM defines the spectral resolution, but if those values are stated anywhere, I missed them.

*Line 93:* This paragraph should conclude with a reference to the later sections in which the non-ideal response of the ALI significantly affects the analysis (to motivate the detailed discussion of this feature, and also to guide a reader who is particularly interested in this aspect of the work).

*Lines 101-102:* "external source each pixel measured" should be "external source for each pixel measured."

*Lines 103-104:* Here and elsewhere, the set of images used to characterize the dark behavior is called "large" – how large? And how was the most appropriate large number chosen?

*Line 111:* "Bad pixels" are mentioned in this paragraph (and defined in the next). Later (in Figure 4), some visual evidence is provided that the number of "bad pixels" is a small fraction of the total image, but how small is it (as a fraction of the total number of pixels)?

*Lines 138-139:* "Out-of-field" stray light is a serious problem for many limb scattering sensors (usually more serious than "internal" stray light). Is that true for ALI as well? The text says that this is "mitigated with careful baffling" – can you say more about how well this mitigation works, and how its effectiveness was assessed? (If some of these instrument-related questions are addressed more completely elsewhere, then adding a specific reference would be helpful – and I particularly apologize for not reviewing Letros et al., 2024, due to lack of access!)

*Table 1*: I recommend including the estimated albedo for each scan as another column of data. That would make it immediately clear why Scan 3 includes more highly polarized observations than the others (lack of underlying clouds, which suppresses the amount of multiple scattering present).

*Line 155:* This paragraph says that the scans presented represent "reasonably nominal conditions" for ALI. Besides the gondola being "relatively stable," how are "reasonably nominal conditions" defined? I assume that the statement about stability refers to the attitude of the sensor – how is this assessed? And do variations in pointing during the integration time of the measurements contribute significantly to the effective altitude resolution of the measurement (or affect measurement quality in other ways)?

*Lines 184-185:* The definition of the D matrix should be clarified. Would it be fair to call this a Tikhonov regularization term? Stating that it "restricts elements of the optimization from changing too much" should be reworded - I would describe it as being meant to retard the change in the state vector from one iteration to the next in the early phases of the retrieval. (This is reasonable in a Rodgers-type optimal estimation scheme for multiple reasons: An a-priori profile that differs greatly from the true profile may cause the initial retrieval step to move much too far, or in the wrong direction, for example.)

*Line 208:* "… as discussed later" – this should include a reference to the section where the material appears.

*Figure 6:* The text provided in Section 4.1 suggests that the albedo estimation process used in this work is new (or at least significantly modified, relative to earlier approaches). But many of the details of the algorithm used are described in the figure caption. This makes the figure caption too long, and fills it with information that isn't directly related to the figure, without providing enough detail about the method itself. I recommend moving most of this caption into the regular text of the paper, and perhaps expanding it (or adding references) to clarify how it works.

*Line 226:* "… atmosphere of known true state albedo of 0.6." What is the "true state albedo" – the actual albedo of the underlying surface? An effective reflectivity (combining the influences of the surface, clouds, aerosols, etc.)? Or something else? This is related to the following note.

*Line 228:* "… 0.654 was found, which we consider to be a reasonable estimation of the 0.6 true state." My initial reaction is that this agreement (worse than 0.05, or nearly 10% relative error) is not especially good… but maybe that's unfair. Does the simulated data used in this example include noise, biases, etc. that are meant to mimic the performance of ALI? Does the stated value (0.654) represent an "effective reflectivity" that cannot be expected to perfectly match the "true" value (0.6), for the reason noted in the line 226 note (or for some other reason)? And maybe most importantly, how much does an error in retrieved albedo matter for the retrieval of the aerosol properties? (As noted elsewhere in the text, tangent height normalization often significantly reduces the sensitivity of limb scattering retrievals to uncertainty about the brightness of the scene.)

*Lines 237-238:* This sentence contains a particularly confusing reference to cloud effects: "… relative changes in polarized light can be used as a metric to determine if limb-scattered signal was influenced by cloud or not." Are we talking about the

"influence" of clouds in the underlying scene here? That's my interpretation, but maybe the statement also applies to line-of-sight clouds? This should be clarified.

*Line 268:* The reference "Bass and et al, 2010" appears here. That appears to be the only reference that uses "et al." rather than listing the full author list – why? And calling it "and et al." seems redundant.

*Line 280:* "standard divinations" should be "deviations." And it would be helpful to explain what "prototyping the algorithm" means. I assume this involved experimenting with various settings – you settled on these particular settings for some reason, but how much did the particular selections that you made matter, in the end?

*Line 288:* "… the $\theta$ state is reasonably insensitive" – this should be quantified.

*Line 310:* Here (and in the caption of Figure 7), the text refers to a "stark" change in DoP behavior (for cloud identification). The figure provides some visual demonstrations that these can appear very obviously in the profile, but how "stark" must a change be to trigger identification of a feature as a cloud?

*Line 313:* "… known form the true state" – should be "from."

*Lines 375-376:* "… variance of the median radius is … selected to be 0.01." and "The scalar width has this variance set to 0.0001." Shouldn't these values have units? I may be confused about what these definitions mean… but again, how much do they matter in the observed behavior of the retrievals?

*Table 3:* How were these listed number density variance values selected, and how do they affect the retrieval? As mentioned earlier, I'm particularly curious how the solution might be affected when the a-priori number density profile differs from the true atmosphere by a factor of 10 or more.

*Lines 405-406:* "However, of note this a-priori $N$ profile is not constructed from any specific knowledge of the aerosol to be retrieved." What does this mean? I guess the a-priori profile is some kind of climatology, or …?

*Figure 10:* This caption ends with the statement that "This small error primarily results from the horizontal averaging." For a small error, maybe I shouldn't quibble, but I have to ask: What quantity is being averaged horizontally, and how? Is this averaging discussed elsewhere in the paper, and is it obvious how you determined that this averaging is the cause of the observed error?

*Figure 12:* The properties of the bimodal distribution are shown in the figure, but do you have a reference for how these particular properties were selected?

*Lines 459-460:* "This is a similar approach to the standard retrieval approach of OSIRIS and OMPS (Rieger et al., 2019; Taha et al., 2021)." For the latter case, the assumed aerosol size distribution is not log-normal. As stated at the start of the review, I appreciate the experiments with bimodal log-normal distributions to test the approach, but were any non-log-normal experiments also done?

*Lines 465:* "… yields respectable agreement overall." How is this defined? (With the extinction plots presented on a logarithmic scale, it's difficult to read the percentage error well.)

*Line 503:* "… atmospheric DoP can be well retrieved" – this should be quantified.

*Line 515:* "… but we this is a point" – should be "but this is a point."

*Line 519:* "… extinction in very good agreement" – this should be quantified.

*Line 521:* "… showed an overestimation of aerosol extinction" – this should be quantified.

*Line 503:* Is listing data as "available upon request" adequate? I understand that public release of the full set of ALI measurements may not be possible. But inclusion of the data used and illustrated in this study (as a public "supplement" file) seems like a reasonable expectation, which I thought had become a fairly standard practice (for data that is not "officially released" and archived elsewhere).

---

## Referee Comment (RC3)

**Referee report to the "The Aerosol Limb Imager: Multi-spectral Polarimetric Observations of Stratospheric Aerosol" manuscript by Daniel Letros et al.**

The manuscript describes a retrieval algorithm for the new Aerosol Limb Imager (ALI) instrument and presents some results from the synthetic retrievals and from three example measurements made by the instrument. Although the measurement concept of ALI is similar to that used by the upcoming ALTIUS mission of ESA, it offers an unique feature of measuring the polarization state of the limb-scatter radiance. Authors did a great job showing how this feature of ALI can be used to detect contamination by clouds, which has always been an issue for limb-scatter aerosol retrievals. Less impressive are, however, the results from the aerosol retrieval itself. Here, I got an impression that the polarization-sensitive measurements make the instrument useless for the aerosol retrieval. This is not an issue for the scientific significance of the paper but the authors, especially PIs of the project, should think about if they really want to provide this impression to the scientific community. Unfortunately, the presentation of the results is elaborated quite poor and needs to be improved. To improve the message of the paper authors need to quantify the required conditions for the retrieval of reasonable aerosol extinction coefficient profiles from real measurements. A careful proof-read of the paper is needed to correct typos, extra or missing words etc.

**General comments**

- Date and time of the ALI measurements and those of the collocated reference measurements are not provided. This make impossible to understand which aerosol conditions were investigated. Furthermore, no information about collocation criteria is given.

- Simulation results do not look representative enough. The aerosol extinction above 30 km was set to zero and no attempts were made to check what happens if a realistic aerosol distribution at high altitudes is included. It is not clear for which surface albedo the simulations are done and what happens if albedo changes. How the estimations of the surface albedo are affected by the presence of the aerosol above 30 km? Are the simulation results remain the same if an aerosol profile for different aerosol loading conditions is used? Logarithmic plots and absence of relative difference plots make evaluation of the retrieval quality nearly impossible. If I understand it correctly, the synthetic retrievals were done without adding measurement noise to the simulated spectra. How well the retrieval chain works if measurement noise is added (I mean here using noisy simulated spectra, not only adding the noise covariance matrix into the retrieval)? How the aerosol parameters for the exercise with bimodal aerosol PSD were set, where comes the information about the used parameters from? Are they realistic? Some quantification of the results in the case of the bimodal PSD is needed, i.e. some realistic bimodal distributions corresponding to different aerosol loading conditions have to be found and used for simulations. Comparison of the

mode radius and width from the unimodal and bimodal distributions does not make much sense. A comparison of the effective radii might be more useful.

- Comparison with measurement data is difficult to evaluate without having the relative difference plots. It is not clear for which aerosol conditions the comparison with reference data was made, this information is extremely difficult to derive from the logarithmic plots. The overall conclusion from the comparison seems to be that only one of the three measurements produces reasonable results although data from both OSIRIS and OMPS-LP work well in all three cases. Is it a general issue of the applied technique? Is the technique then useful at all? A more detailed discussion of the usability of ALI results needs to be done at this point.

**Detailed comments**

- Line 6 and 34: At this point it is unclear what the term "scalar width" means.

- Introduction lacks some information about how the ALI instrument is ranged with respect to the previous, present and planned space-borne aerosol instruments.

- Section 2: Some technical data of the instrument need to be provided, e.g. vertical/horizontal resolution and sampling, FOV range, typical exposure time etc.

- Lines 51 - 52: "The SPS also contains one linear polarizer after the LCR and another after the AOTF to further refine the polarized image" – There is a polarizer before the AOTF and AOTF itself passes one polarization direction through. Please explain shortly why an additional refinement is needed.

- Figure 1 is quite dark and depending on the monitor and illumination difficult to see. Please light up the dark parts of the figure.

- Figure 2, panel (b): the abbreviation "DE" is not defined in the caption.

- Line 69: "The spectral bandpass of ALI for a tuned frequency is taken as the full-width half-max (FWHM) of the diffraction response." – it looks like, here, you use the "diffraction response" term to describe the same function as shown in the panel (b) of Fig. 2 an referred to as the "spectral response". In line 75 you write, however "spectral response is calculated by integrating the measured diffraction responses" which suggest different meaning of these two terms. Please clarify.

- Figure 3 caption: "LCR on state" – here and throughout the text, it would be less confusing is you wrote "on" in the quotation marks. The same is for "off".

- Line 99: please give some details to explain what the "calibrated broadband integrating sphere of known (randomly polarized) spectrum" is.

- Line 100: "Furthermore, since they reproduce the spatially flat and full-field conditions of the integrating sphere, they can be used to relate the ALI measurements to the external source each pixel measured."– Does "spatially flat" means homogeneous of a flat shape of the source? It is not quite clear how to get a response to an inhomogeneous source signal if you have a response to a homogeneous full-field signal. Please explain. Maybe I misinterpret what you want to say with this sentence, in this case please rephrase for more clarity.

- Line 128: Are there any investigations of the temporal stability of the obtained calibration parameters and possible dependence on the ambient temperature and pressure, strength of the illumination etc. ?

- Figure 3 caption: "White dots in this image indicate the bad pixels as determined by the dark correction." – I do not see any white dots in panel (a), what do the black dots mean?

- Figure 5 caption: The sentence "The radiance profiles of (b, c, e, f, h, i) are constructed by following Section 2.3 to convert images into photons/s/cm2 /sr from DN. Then following Section 2.1 to obtain units of photons/s/cm2 /sr/nm." should be moved to the main text.

- Figure 5 caption: "the solar irradiance produced by a radiative transfer forward model" – solar irradiance is not produced by radiative transfer models.

- Line 206: "a regularization matrix in place of $S_a^{-1}$" – $S_a^{-1}$ is also a regularization matrix, do you mean Twomey-Tikhonov regularization matrix, i.e. a smoothing constraint here?

- Line 208: "measurement and state vectors of large dynamic range which tends to produce ill-conditioned inversions." – the ill conditioning of the inverse problem is not caused by the large dynamic range of the measurement and state vectors. Its reasons are rather a dense vertical sampling, insensitivity of the retrieval to certain vertical ranges and correlation between different parameters.

- Line 214 - 217: In most previous aerosol retrievals the albedo and aerosol retrieval were run alternatively within an iterative process allowing the retrieved albedo to adjust to the retrieved aerosol profile and vise versa. In this retrieval, the albedo is retrieved only once before the aerosol retrieval. Additional investigations need to be done to show that the retrieved albedo does not depend significantly on the a priori aerosol profile used for the albedo retrieval.

- Line 221 - 222: I do not expect that the influence of aerosol at 33 – 34 km is minimal. This statement has to be proven using measurement data, e.g. GloSSAC.

- Sect. 4.1 please provide the aerosol profile used for albedo retrievals. Dependence of the albedo retrieval on the assumed aerosol profile needs to be investigated.

- Lines 245 - 246: "Therefore, we also present an approach to retrieve the Stokes parameters of the atmosphere in the Stokes basis of ALI ..." – Please explain what "the Stokes basis of ALI" means.

- Eq. (7): What happened to U and V components. Were they just neglected? Please clarify.

- Lines 267: "...which directly correspond to tangent altitudes at the time the observation was taken" – What do you want to highlight with "at the time the observation was taken", are the tangent heights measured not at the same time?

- Lines 300 - 302: "In this example $\Theta$ received little to no action by the inversion to adjust it form the a-priori state. It is sensible to simply not include $\Theta$ as a property in x and just rely on the a-priori values in the forward modeling." – Is it a common behavior or does it just happen occasionally?

- Line 310: "... we convolve the smoothed DoP with a central difference impulse response.." – please explain what is the "a central difference impulse response" or give a reference.

- Fig. 7: Please provide an illustration what happens if there is a strong aerosol level with larger particles in place of the cloud layer.

- Line 333: "At each scan, the true state of the atmosphere includes GloSSAC aerosol but unlike the exercise of Fig. 7 and Fig. 8 no ice layer is included." – please show the aerosol profile used in the retrieval (if you like in the Supplement) and prove the results are the same for a different aerosol loading scenario.

- Sect. 4.2.4: It would be interesting to have a larger statistics about the working and non-working DoP retrievals.

- Line 349: "retrieved along side a scalar width" – you still haven't properly defined what scalar width means.

- Line 369: "as the finest resolution $\mathbf{A}$ produced" – What does "$\mathbf{A}$" mean here?

- Lines 368 - 370: Resolution might be limited by the forward model grid but not determined by it. It is mainly determined by the instrument FOV and sampling and also applied regularization. What you mean here is the sampling.

- Line 371: "we do not employ regularization" – Using $S_a$ is also a regularization.

- Lines 391 - 394: The description looks like the Levenberg-Marquard method, why do not you use the name, is the any significant difference?

- Sect. 4.3.1: Please show the GloSSAC aerosol profile used for the study. Why was only one profile used? Dependence on the true aerosol profile might be significant. Different aerosol loading conditions need to be investigated.

- Line 400: Please provide the illustration what happens if you do not set the aerosol profile above 30 km to zero.

- Figure 10: Is it correct that the true surface albedo is not changed here?

- Line 416: What happens if a priori and true widths are the same but the width is included into the retrieval?

- Figs. 10 - 13, 15, and 16 must include relative differences.

- Line 433: "We wish to emphasize that the limb measurements of ALI are polarized, and we speculate that this polarized content contains useful information about the aerosol phase scattering matrices - which is of course influenced by the particle sizes" - also non-polarized measurements contain information on the particle sizes.

- Sect. 4.3.1: Authors found a strong dependence of the results on the presence of the second mode for highly polarized measurements, the question is if polarized measurements are really an advantage or a drawback. This should be discussed in more details in the text.

- Lines 448 - 451: "With respect to the simulations done in the exercise surrounding Fig. 9, we find almost all behavior regarding Fig. 14 to be expected including: the relative balance between horizontal and vertical polarizations for all three scans, the spectral regions expected to fail given the polarimetric response of ALI in Scan 1 and Scan 3, and the relative magnitude of the DoP for Scan 1 and Scan 2." – this discussion is difficult to follow, please write in more details what exactly you expect from the discussion around Fig. 9 and what we see in this respect in Fig. 14.

- Lines 448 - 451: "The simulation of this geometry done for Fig. 9 produced a true state DoP approximately ranging between 0.3 - 0.4" – please provide corresponding plots, if you prefer they can be in the Supplement.

- Line 473: "... of interest is the profile of r which indicates a layer of larger particles at approximately 22.5 km..." – please compare this results with SAGE III data and make a statement about the agreement.

- Lines 485 - 488: A large disagreement for ALI and a good agreement for other instruments means, in my opinion, that the instrument concept where the polarized measurements are used is not optimal for the retrieval of the aerosol extinction coefficient. Please discuss this issue.

**Technical corrections**

- Line 33 and throughout the text: "dependant" $\longrightarrow$ "dependent"

- Figure 1 caption: "comprises of one off-axis" ⟶ "comprises one off-axis"

- Figure 2 caption: "shown in black" ⟶ "is shown in black"

- Figure 3 caption: "response of ALI (Letros et al., 2024) shown as" ⟶ "response of ALI (Letros et al., 2024) is shown as"

- Line 117: "be used to constructed" ⟶ " be used to construct"

- Line 128: "conversion to from DN into" ⟶ "conversion from DN into"

- Line 170: "Here we discuses the concepts" ⟶ "Here we discus the concepts"

- Line 175: "and marks the lower altitude limit to retrieve" ⟶ "and mark the lower altitude limit to retrieve"

- Line 347: "to emphasise that the in the context" ⟶ "to emphasise that in the context"

- Line 347: "criteria we set is to retrieve" ⟶ "criterion we set is to retrieve"

- Line 349: "their is" ⟶ "There is"

- Line 364: "ceiling on of the radiance" ⟶ "ceiling on the radiance"

- Line 396: "summary of retrieval result" ⟶ "summary of retrieval results"

- Line 407: "by retrieving a N and r profile" ⟶ "by retrieving N and r profiles"

- Line 413: "by the biased caused" – do you mean "by the biases caused"?

- Line 417: "shows the behaviour" ⟶ "show the behaviour"

- Line 515: "but we this is a point" ⟶ "but this is a point"

---

## Author Comment (AC1)

**Author (Daniel Letros) responses to the referee comments is shown in blue.**

**Our entire team would also like to thank you for your feedback. Your efforts have helped to improve our work and we are appreciative!**

The authors have put together a very impressive paper to provide a brief description of their instrument (the Aerosol Limb Imager, aka ALI) and its aerosol retrieval algorithm. Herein, the authors provide a thorough presentation of their algorithm and discuss some of its limitations in a comprehensive manner. The ALI instrument takes advantage of limb scatter measurements at multiple wavelengths for 2 polarization states. This information content enables the authors to retrieve aerosol radius and number density estimates as well as a rather rudimentary estimate of the distribution width, which is assumed (forced?) to be constant throughout the profile. I must reiterate that overall I am pleased with this paper. It is well written (up until sec- tion 4.3 where typos became more plentiful as did missing articles), logically organized, easily read/followed, and generally well supported with ample use of figures. I believe this paper will make a important contribution to the scientific literature, that it fits well within the scope of AMT, and should be published pending some revisions.

**1. General comment:** The authors used SAGE III/ISS, OMPS, and OSIRIS data to evaluate their derived extinctions, but nowhere do they tell the reader what profiles were used (e.g., profile or event number), do not tell the reader where these profiles were collected (e.g., lat/lon of the satellite observations), and they fail to inform the reader of when these profiles were collected. All of this information is necessary for interpreting these intercomparisons (some of this may explain the differences they observed between the satellite instruments and ALI). This information should be added before publication.

We have made a supplemental PDF which contains information tables and a map showing the coincidence of the ALI measurements with the other three instruments. We have also made references to the supplemental document in both the caption of Table 1 and elsewhere in the paper.

**2. General:** The authors failed to inform the reader of where ALI was sampling. The authors stated in section 3 that "ALI was situated on the balloon gondola and orientated such that when the gondola is flat and level, the highest lines of sight (top pixels of the ALI detector) would be horizontal and with tangent locations on the instrument itself " (emphasis added). The question is: where is the bulk of the scattering coming from? It's certainly not the "tangent point" (i.e., at the instrument itself). Maybe the distance is negligible, but the reader probably does not know that. This information should be included before publication.

We consider the location of the ALI scans to be at the mean tangent position of each ALI scan. We indicate this in the supplemental document that was made to address your comment above. In addition we have swapped the lat/lon information of Table 1 from being the position of the gondola (which is relatively unimportant information within the paper) to be that of the mean tangent positions of the measurements. The caption of Table 1 has also been updated to reflect this.

**3. General:** The authors miss an opportunity to compare their radius and number density estimates with those released by the SAGE team. If there were coincident OPC data available I would recommend they use that, but since the SAGE data is all that's available I must strongly recommend the authors include that data in their analysis before publication.

We have added this comparison with discussion to the paper.

**4. General/Figures:** The figures could benefit from being larger, which would enlarge the fontsize of the axes and make them easier to read. Please consider making this change.

Further efforts have been made to make the figures more readable and properly formatted in the final submission.

**5. Page 6, line 122:** The authors claim that the error is "...on the order of the square root of the DN values..." Maybe I am being too precise, but the FWHM in Fig. looks to be ≈40, which corresponds to a DN of 1600 (i.e., $40^2$) whereas panels (a) and (b) indicate DN on the order of 8000. Would the authors please clarify their meaning here?

Your numbers are approximately correct. This statement was just meant to be a digestible one indicating that this methodology is not introducing significant error in the analysis. We have reworded it for this explicit intent:

Figure 4 shows an example of synthetic image construction compared against an actual ALI image. As this example shows, the synthetic image provides a faithful recreation of the actual ALI measurement. The error of this reproduction, as demonstrated by the histogram FWHM of 33 DN, is approximately 0.4% of the mean of the image signal.

**6. Fig. 4:** What is the black dot in panel (a)? That corresponds to the large white dot in panel (b), so I assume it is a bad pixel, but all other bad pixels are white. Would the authors please clarify?

White dots are indeed bad (non-ideal) pixels. The lack of any non-ideal pixel in (a) with respect to (b) comes from the level of calibration applied to the image. (a) is an ALI image of the integrating sphere which has only dark correction applied (i.e. only pixels which are identified as non-ideal under non-illuminated conditions are white). (b) is a synthetic reproduction of (a) which includes non-ideal pixels under both illuminated and non-illuminated conditions. The pixels identified as non-ideal under illumination (such as the block dot you mention) are determined from images like that of the one shown in (a).

The caption of Figure 4. does allude to this, but it is not as clear as it should be. We have adjust the wording:

Figure 4. Example comparison of synthetic image construction for ALI correction. (a) A single ALI image of the calibrated integrating sphere. AOTF is tuned to diffract 1450 nm, the LCR is off, and an exposure time of 0.450 seconds is used in this example. Dark correction has been applied. White dots in this image indicate the bad pixels as determined only by the dark correction. (b) A synthetic reproduction of the real image shown in the left plot constructed from the database of calibration coefficients. White (bad) pixels seen in this image are determined as pixels with non-ideal responses in both dark and illuminated conditions. These pixels are discarded. (c) A histogram of all (non-bad) pixel values after the synthetic (middle) image is subtracted from the real (left) image.

**7. Page 7, lin 143:** The authors refer to ALI's viewing geometry "...when the gondola is flat and level..." I have two questions regarding this:

(a) How often is the gondola level and flat? I assume there is some stabilization utilized, but it is never explicitly mentioned.

(b) Is this orientation monitored to allow correction? Was a correction applied? How does this variation impact the view geometry and the results?

The initial statement of "...when the gondola is flat and level..." is meant to indicate the fixed orientation of ALI mounted on the gondola. Of course, the gondola itself is not perfectly flat and level during flight and has its own time-varying attitude. To answer both (a) and (b) together: the gondola is pointed and its momentum can be transferred to the helium balloon giving stability to the instruments. The orientation of the gondola is tracked by IMUs which allow us to reconstruct each ALI line of sight of each image. In selecting the three scans used in the paper, attention was paid to the variation of the gondola during image exposures to minimize what would manifest as spatial blurring.

As alluded to in the paper, Scan 1 was almost completely stable in attitude over the exposures but the other two scans also have minimal changes (i.e. changes in orientation exceeding the uncertainty of the

IMUs). To provide an idea of the gondola stability you will find a plot of the flight reconstruction below. Scan 2 (taken at about the 7 hour mark in the attached plot) is the worse of the three used scans - although again we will emphasize it is still relatively stable with < 0.1° change in pitch over each image acquisition. While we did not present other retrievals of less stable scans in the paper, when studying those we binned detector rows together by tangent altitude as a strategy to accommodate the lost of spatial resolution of the measurements within the retrieval.

[Figure]

Figure 1: Reconstruction of ALI position and attitude on the flight gondola during the Timmins 2022 flight.

We refrain from discussion around the gondola attitude and flight reconstruction within the paper for brevity, as we hope to maintain scope around the retrieval algorithm and ALI science demonstration. However, for other readers we have now appended to the first paragraph of Section 3:

[...] At this point in the flight the gondola was steered to maintain a solar azimuth angle (SAA) of 60°. During the flight the gondola position and orientation is recorded which allows reconstruction of all the ALI lines of sight in each image.

... and these in the third paragraph of Section 3:

[...] The balloon gondola was relatively stable for this scan compared to most others taken during the flight (gondola attitude within IMU error during exposures), [...]

[...] The other two scans we select present more difficult observation conditions both in terms of relative gondola stability (although there was still minimal attitude change during exposures, i.e < 0.1° change in pitch over each image acquisition) and observational conditions.

**8. Fig. 12:** I wonder if the authors sell themselves short on this figure. You cannot solve for bimodal distributions, but you may be able to reasonably infer the effective radius, which would resolve some of the "bimodal issues" the authors allude to throughout sections 5/6 as well as provide a more robust number for use in models. That said, the constant width value may skew re. I suggest the authors include another panel in this figure to evaluate re.

This is an excellent thought. We provide you the effective radius profiles of the retrievals in Fig 12 and Fig 13 below.

[Figure]

[Figure]

(a) Exercise of Figure 12 (geometry of Scan 3)

(b) Exercise of Figure 13 (geometry of Scan 1)

Figure 2: Aerosol effective radius of bimodal simulations.

In the specific case of simulation we do not think that they add any new insight or information beyond what is already shown. A core finding we are attempting to convey in the paper is that polarized limb measurements are more sensitive to aerosol particle size distributions. Unimodal treatment of a bimodal distribution can be insufficient. To this end, failing to retrieve the correct extinction in the exercise of Figure 12 (using the geometry of Scan 3 where the atmospheric degree of polarization is higher than Scan 1) shows this shortcoming. Likewise, simulating the same aerosol state but in a less polarized atmosphere (geometry of Scan 1) in Figure 13 yields noticeable improvement. The effective radius profiles ultimately reflect the same improvement the extinction profiles do. We also attempt to convey that a similar looking improvement between Scan 1 and Scan 3 in the real retrievals (Fig 16 and Fig 18) is observed, although with a much less realistic state in Scan 3 as you note below.

**9. Page 23, line 460:** The authors state "In this simpler retrieval only N is adjusted in x such that we arrive at an aerosol extinction directly retrieved at 750 nm." As written, it sounds like the authors are iterating to match an extinction, when I think they are iterating to match the radiance (like OMPS and OSIRIS). Would you please clarify?

We have reworded this sentence (along with other minor rewording later in the paragraph) in an attempt to be more clear :

In this simpler retrieval only $N$ is retrieved in the state vector $\vec{x}$, with only the 750 nm radiance constructing $\vec{y}$.

**10. Page 24, line 465:** The authors state "Retrieving extinction at only 750 nm yields respectable..." (emphasis added). "Respectable" is ambiguous, please quantify and clarify. Addressed in 12.

**11. Page 24, line 469:** The authors state "...shows a fairly ideal retrieval." What is meant by "fairly ideal"? I have no idea what is meant. Can the authors be quantitative or provide a metric for the reader to gauge "idealness"? Addressed in 12.

**12. Page 24: line 470:** The authors state "...the retrieved extinction well represents the ex tinction profile of all three comparison instruments." Again, "well represents" is ambiguous. The figures in reference (15, 16, 17) are plotted on log scales, which makes quantitative evaluation challenging. Readers would greatly

benefit from a percent difference plot (or a ratio plot where the 3 comparison instruments are divided by ALI's extinction). This would convey a wealth of information to the reader. Would the authors please include these plots?

Addressing points 10, 11, and 12 together: We have added percentage difference plots to the results and added wording to quantify such statements.

**13. Fig. 18:** I think the authors are overly optimistic in the interpretation of panels (a) and (b). There is no way the number density is so large (it's at least an order of magnitude too high, even after a major eruption) and the sensitivity of their instrument (at the designated wavelengths) to particles with radius of 0.04 µm is questionable. I would suggest that if the authors want readers to take panel (b) seriously then they should provide more support (maybe show scattering intensity at the current scattering angle for the wavelengths in question as a function of particle size?).

We do not expect the state of this retrieval to be taken seriously in an absolute sense - especially so within scope of comparison to the Scan 1 retrieval and the newly added SAGE particle size comparison. In line 492 I state "We speculate that this is another manifestation of the retrieval trying to optimize a unimodal distribution to match the polarized $\vec{y}$ produced by a more complicated aerosol population." indicating we don't take it seriously ourselves. The discussion surrounding this part of the paper is attempting to highlight the positive aspect that the retrieved profiles are similar in shape to Scan 1, and further demonstrating this polarization point.

**14. Page 29, line 522:** Regarding the sentence starting with "However, the disagreement of these two...": The authors put all the blame on a bimodal distribution, which may or may not be the case. However, I don't think this is supported and I am unsure of what the authors are trying to communicate within the last 2 sentences of this paragraph. Are they suggesting that the instrument is seeing different atmospheres in the various scans and scan 2 observed aerosol with a bimodal distribution? Are they suggesting that the instrument is seeing the same atmosphere, but the profiles are different because a bimodal distribution has more impact at some scattering angles than others? I don't know. Would the authors please clarify?

We have clarified the wording:
[...]Scan 2 and Scan 3 showed an overestimation of aerosol extinction with respect to SAGE III, OMPS, and OSIRIS. However, the disagreement of these two scans can potentially be explained by the affect of using a unimodal distribution to represent more complex aerosol under polarized limb measurement conditions. We speculate that the same underlying retrieval behaviour seen between the simulated exercises of Fig. 12 and Fig. 13 may be occurring in the real retrievals as well. Supporting this statement is the relative improvement in the quality Scan 1 exhibits which is also replicated in simulation under this case.

---

## Author Comment (AC2)

**Author (Daniel Letros) responses to the referee comments is shown in blue.**

**We would like to extend our appreciation to you for your efforts in giving us this feedback. It has made the presentation of our work better. Thank you!**

**General comments on Letros et al. [2025]:**

This is a well-executed paper that presents analysis of data taken by the ALI instrument, which records polarized radiance spectra. The value of these measurements is demonstrated through both sensitivity studies and analysis of selected ALI data. A particularly useful capability provided by radiance information is the ability to distinguish between clouds and aerosols along the line of sight.

I have few criticisms of the work presented, and particularly appreciate the degree of testing presented for the algorithm approach taken (which is clearly designed to test its limits, rather than primarily to show its performance in the best possible light). Error analysis is thoughtful and thorough. Some structural suggestions and clarification questions follow, but I have a positive opinion of the paper, and would be glad to see it published.

**Detailed comments:**

*Abstract, line 4:* "Tangent altitudes which have signal contaminated by clouds" should be defined more clearly. Throughout the paper, similar wording is used, and the authors' interpretation of this language appears to be consistent: This language is meant to describe tangent altitudes for which a cloud appears along the line of sight. But that's not the only possible interpretation: For example, limb scattering observations can (and frequently are) "contaminated" by upwelling radiation from a broad area below the line of sight, for which the locations and properties of the clouds are not well known. That is a distinct problem from interpreting measurements with clouds along the line of sight, and it isn't the focus of this paper, which the abstract should make clearer.

Yes you are right and our wording should be clearer. In general we are pursuing a quantitative distinction of lines of sight that one would not want to include in an aerosol retrieval - which at a minimum are lines of sight looking through clouds. However, we are aware of the upwelling cloud contamination in practice. To combat this for the flight retrieval efforts in the paper, we limit the retrieval when the radiance profiles see a distinct change in DoP - which (as Figure 7 shows) can be higher than the actual height of the cloud. Exactly how robust of a strategy that is to avoid *all* cloud contamination remains uncertain to us for now but we think it is very fair to say it is not full proof.
We have attempted to clarify the definition of "contaminated by clouds" to be related to the depolarization in the abstract as well as elsewhere in the paper.

*Abstract, lines 14-15:+* The abstract ends by declaring "good agreement" between the aerosol extinction coefficients derived here and coincident data (from SAGE III, OMPS and OSIRIS). It would be more useful to make this statement quantitative, or at least indicate what criteria were used to conclude that the agreement is "good." quantified wording.

The statement about particle size information would also benefit from clarification: A short summary of the particle size distribution assumptions made and the relative quality of the various retrieved properties would help a reader who is reading the abstract before deciding whether to proceed further. Added to the abstract concerning the new SAGE particle size comparison. We believe it addresses this concern.

*Line 47:* "purposes of the SPS is" – should be "are" Corrected.

*Figure 2:* The spectral response functions shown in parts (a) and (b) appear to be fairly Gaussian – it would be useful to include an estimate of the FWHM (or some other indication of their width). The caption

mentions that the FWHM defines the spectral resolution, but if those values are stated anywhere, I missed them.

The spectral resolution shown in Figure 2(d) is directly the FWHM of the spectral response functions. We note this in the text of the first paragraph of Section 2.1 - however we understand your confusion. In the text we call it spectral bandpass and not resolution as (d) is labelled. We have clarified the wording of the text and also noted the FWHM relation in the caption of Figure 2(d).

*Line 93:* This paragraph should conclude with a reference to the later sections in which the non-ideal response of the ALI significantly affects the analysis (to motivate the detailed discussion of this feature, and also to guide a reader who is particularly interested in this aspect of the work). Very good thought. Implemented.

*Lines 101-102:* "external source each pixel measured" should be "external source for each pixel measured." Corrected.

*Lines 103-104:* Here and elsewhere, the set of images used to characterize the dark behavior is called "large" – how large? And how was the most appropriate large number chosen?

Of course, to characterize the dark (or any other) behaviour, the more images collected the better. Generally we had no specific chosen number of images to collect for any test, and the dark characterization for instance had a variance in the number of images collected at each exposure time. The efficacy of our calibrations (which included if we needed more image collection) was determined by analysis. I.e. the dark calibration was validated by looking at the histograms of randomly selected dark images and seeing the deviation of the mean from zero. Additionally, Figure 4 of the paper exemplifies our evaluation of integrating sphere calibrations.

However, to loosely quantify "large" here we have noted hundreds of images at line 103-104 - which was typical of our calibration work.

*Line 111:* "Bad pixels" are mentioned in this paragraph (and defined in the next). Later (in Figure 4), some visual evidence is provided that the number of "bad pixels" is a small fraction of the total image, but how small is it (as a fraction of the total number of pixels)?

We consider 7289 pixels out of the 327680 total pixels (2.22%) to be non-ideal and exclude them from any scientific analysis.

*Lines 138-139:* "Out-of-field" stray light is a serious problem for many limb scattering sensors (usually more serious than "internal" stray light). Is that true for ALI as well? The text says that this is "mitigated with careful baffling" – can you say more about how well this mitigation works, and how its effectiveness was assessed? (If some of these instrument-related questions are addressed more completely elsewhere, then adding a specific reference would be helpful – and I particularly apologize for not reviewing Letros et al., 2024, due to lack of access!)

This is an excellent question, and we do not have this granularity of the ALI design published anywhere currently. The short answer is we do not think out-of-field stray light plays a significant role in ALI performance (although it is of course always a concern of any limb instrument). Our confidence for this comes from analytical study of the design (with commercial optical software) and the lack of any significant evidence of it seen in lab conditions. Furthermore - although obviously not controlled conditions for analysis - we have seen no evidence to suspect it being in the flight data either.

To elaborate a little further, stray light was studied in the lab in two ways: (1) by systematically moving our integrating sphere all around the ALI optical enclosure and imaging with the AOTF off, and (2) rotating the ALI field of view in front of a collimator and imaging with the AOTF on. Test (1) is an excellent test

for general light leaks like from gaps in the optical housing. After correction of these images to measure the stray light we observed signal (at worse) on the order of tens of DN. These images would otherwise well saturate the detector (16k+ DN) if the AOTF were on and ALI was looking at the source. However, (1) is not a good test for out-of-field stray as the field angles of the source are not well controlled and the AOTF is off. Test (2) accomplishes this, and as we systematically rotated the collimated beam out of the intended ALI field of view we saw dark images - indicating the baffling was working as designed.

However for full transparency, physical limitations of our lab equipment made rotating ALI in front of the collimator very impractical for steeper angles (i.e. $> 5°$ outside of our field of view) - and unfortunately the limitation is in the vertical (altitude) direction which is of course more important to the science. It is at this stage of evaluation that our confidence relies on the analytical optical design and not on specific testing. Hence, we will always have trouble giving a definitive "no we do not have stray light" but, again, we do not consider it an issue with ALI.

*Table 1:* I recommend including the estimated albedo for each scan as another column of data. That would make it immediately clear why Scan 3 includes more highly polarized observations than the others (lack of underlying clouds, which suppresses the amount of multiple scattering present).

We refrained from that initially because when Table 1 is provided the albedo is not discussed yet. However, we do think Table 1 can be changed to better provide information about the three scans with the albedo being one aspect. We have altered Table 1 to include more relevant information to the context of the aerosol retrievals (as opposed to the gondola).

*Line 155:* This paragraph says that the scans presented represent "reasonably nominal conditions" for ALI. Besides the gondola being "relatively stable," how are "reasonably nominal conditions" defined? I assume that the statement about stability refers to the attitude of the sensor – how is this assessed? And do variations in pointing during the integration time of the measurements contribute significantly to the effective altitude resolution of the measurement (or affect measurement quality in other ways)?

Largely, "reasonably nominal conditions" here means the stable attitude of the gondola. Although as we allude to in this paragraph and discuss later in the paper, we also mean to imply atmospheric DoP conditions. As for the attitude, it is measured by IMUs which allows the reconstruction of ALI lines of sight post-flight for the retrievals. The main issue (which you also noted) is how the gondola may change attitude during an image exposure which will cause a loss of spatial resolution. We select the scans we do with this in mind to avoid scans which would have significant spatial blurring.

We refrain from discussion around the gondola attitude and flight reconstruction within the paper for brevity, as we hope to maintain scope around the retrieval algorithm and ALI science demonstration. However, for other readers we have now appended to the first paragraph of Section 3:
[...] At this point in the flight the gondola was steered to maintain a solar azimuth angle (SAA) of 60°. During the flight the gondola position and orientation is recorded which allows reconstruction of all the ALI lines of sight in each image.

... and these in the third paragraph of Section 3:
[...] The balloon gondola was relatively stable for this scan compared to most others taken during the flight (gondola attitude within IMU error during exposures), [...]
[...] The other two scans we select present more difficult observation conditions both in terms of relative gondola stability (although there was still minimal attitude change during exposures, i.e $< 0.1°$ change in pitch over each image acquisition) and observational conditions.

*Lines 184-185:* The definition of the D matrix should be clarified. Would it be fair to call this a Tikhonov regularization term? Stating that it "restricts elements of the optimization from changing too much" should be reworded - I would describe it as being meant to retard the change in the state vector from one iteration to the next in the early phases of the retrieval. (This is reasonable in a Rodgers-type optimal estimation

scheme for multiple reasons: An a-priori profile that differs greatly from the true profile may cause the initial retrieval step to move much too far, or in the wrong direction, for example.)

We agree completely with your feedback here and do like your description of retarding change better than what we had used in the paper. We have adjusted the sentence in the paper to now use your wording.

*Line 208:* "... as discussed later" – this should include a reference to the section where the material appears. Reference to Section 4.3 added.

*Figure 6:* The text provided in Section 4.1 suggests that the albedo estimation process used in this work is new (or at least significantly modified, relative to earlier approaches). But many of the details of the algorithm used are described in the figure caption. This makes the figure caption too long, and fills it with information that isn't directly related to the figure, without providing enough detail about the method itself. I recommend moving most of this caption into the regular text of the paper, and perhaps expanding it (or adding references) to clarify how it works.

We have moved much of the caption into the text in an attempt to make this section more elegant.

*Line 226:* "... atmosphere of known true state albedo of 0.6." What is the "true state albedo" – the actual albedo of the underlying surface? An effective reflectivity (combining the influences of the surface, clouds, aerosols, etc.)? Or something else? This is related to the following note.

We understand your confusion here and we should be more clear. The "true state" albedo of 0.6 is meant to indicate the known surface albedo of the simulated atmosphere (the atmosphere we are estimating the albedo of). This atmosphere also includes GloSSAC aerosol (which has no aerosol in the high tangent altitude region we are matching). However, the estimation is meant to quantify your latter definition of effective reflectivity.

To address this confusion we changed mentions of "true state albedo" to "true state surface albedo".

*Line 228:* "... 0.654 was found, which we consider to be a reasonable estimation of the 0.6 true state." My initial reaction is that this agreement (worse than 0.05, or nearly 10% relative error) is not especially good... but maybe that's unfair. Does the simulated data used in this example include noise, biases, etc. that are meant to mimic the performance of ALI? Does the stated value (0.654) represent an "effective reflectivity" that cannot be expected to perfectly match the "true" value (0.6), for the reason noted in the line 226 note (or for some other reason)? And maybe most importantly, how much does an error in retrieved albedo matter for the retrieval of the aerosol properties? (As noted elsewhere in the text, tangent height normalization often significantly reduces the sensitivity of limb scattering retrievals to uncertainty about the brightness of the scene.)

Yes, the simulation is meant to mimic ALI in practice (i.e. the simulated albedo estimation includes noise, bias, etc). Also (as we mentioned in the point above) there is aerosol in the simulated atmosphere, and we are using a Rayleigh (no aerosol) atmosphere to match the up-welling radiation in the 33-34 km tangent altitude range. Given this we would expect the effective albedo we estimate to be higher than just the surface albedo of 0.6.
I have added more robust detail about these points in the last paragraph of Section 4.1.

As for the important aspect - the impact on aerosol retrieval. *Without* high altitude normalization an incorrect albedo will of course introduce a bias in your retrieved aerosol. If you have too low of an albedo, then retrieved aerosol will be biased high to make up for the missing scattered signal. Too high albedo, and the retrieved aerosol will be biased low. *With* normalization, we found in simulation that affect of the albedo is essentially nullified. There are two caveats to that statement though: one) in simulations the physics of the forward model perfectly match (i.e. $\vec{b}'$ of Equation 3.4 in (Rodgers, 2000) is null) and two) in these simulations the aerosol was actually zero within the normalization altitude range.

This second point is the big one which is a limitation of our approach from a balloon platform. While we estimate the albedo to give the retrievals the best fighting chance, we found in practice that a normalization approach was best for the measurements Timmins 2022 flight. We don't consider this surprising as broadly speaking $\vec{b}'$ is not null and normalization *helps* to deal with this. Now being within the stratosphere (unlike a satellite instrument) there is no guarantee that the aerosol is zero at the altitudes we are normalizing by. So $\vec{y}$ is constructed relative to the (potential) aerosol signal within the normalization altitudes. In our approach we make the implicit assumption that it is zero here (as we noted in the paper), but if this assumption is incorrect then the retrieved aerosol will still be biased. If it is biased with respect to other instruments (which we admit ALI seems to be with respect to SAGE, OMPS, and OSIRIS) it is difficult for us to say if it is normalization or another effect, like the bimodal particle size we explore.

To more robustly acknowledge these aspects in the paper, we moved the statement beginning at line 479 (which was meant to make a general acknowledgement that we have limitations) closer to Fig. 15 and increased the detail to include many of the points above.

*Lines 237-238:* This sentence contains a particularly confusing reference to cloud effects: "... relative changes in polarized light can be used as a metric to determine if limb-scattered signal was influenced by cloud or not." Are we talking about the "influence" of clouds in the underlying scene here? That's my interpretation, but maybe the statement also applies to line-of-sight clouds? This should be clarified.

We believe We address this with our response and action to your first comment (*Abstract, line 4*)

*Line 268:* The reference "Bass and et al, 2010" appears here. That appears to be the only reference that uses "et al." rather than listing the full author list – why? And calling it "and et al." seems redundant.

The "and et al." was our mistake in formatting the citation. It is corrected now to just be "Bass et al.". As for the use of "et al." we are happy to make adjustments according to type setting before complete publication.

*Line 280:* "standard divinations" should be "deviations." (Corrected) And it would be helpful to explain what "prototyping the algorithm" means(defined). I assume this involved experimenting with various settings – you settled on these particular settings for some reason, but how much did the particular selections that you made matter, in the end?

Our goal in prototyping the retrieval was to get the correct ballpark settings - in this case the settings of the $\mathbf{S_a}$ matrix. The metric to evaluate the settings was simply if it performed well in simulated exercises with varying true aerosol states. Quantifying how these particular settings matter would be a time intensive study. Qualitatively however, naturally the a-priori covariance will dictate how much certainty to put into the a-priori state. The smaller these covariances are the more iterations the retrieval will need to statistically move away from the a-priori state to one which may better describe $\vec{y}$. Of course, incorrect underlying statistics of the retrieval in general may lead the solution to a false minimum or a very wrong and diverging state. We fully admit that our current algorithm as it is tuned (with regards to settings like the $\mathbf{S_a}$ matrix) may not be strictly optimal.

*Line 288:* "... the $\theta$ state is reasonably insensitive" – this should be quantified.

Now quantified with $< 0.5°$.

*Line 310:* Here (and in the caption of Figure 7), the text refers to a "stark" change in DoP behavior (for cloud identification). The figure provides some visual demonstrations that these can appear very obviously in the profile, but how "stark" must a change be to trigger identification of a feature as a cloud?

The edge detection algorithm has freedom to be tuned (i.e. the amount of smoothing applied or restriction in altitude to even consider a cloud being present) which will adjust what it can detect. Generally though, the change in DoP of the cloud just needs to be distinguishable from the background variance of the DoP profile. So for example in the case of Figure 7 (i), the change caused by the cloud would need to be larger than the background changes (resulting from the noise) in the higher altitudes.

*Line 313:* "... known form the true state" – should be "from." Corrected.

*Lines 375-376:* "... variance of the median radius is ... selected to be 0.01." and "The scalar width has this variance set to 0.0001." Shouldn't these values have units? I may be confused about what these definitions mean... but again, how much do they matter in the observed behavior of the retrievals?

Yes, we should note some units. The variance of the radius should have units of $\mu m^2$(now corrected), and the number density in Table 3 should be noted with $cm^{-6}$. The width is unitless. As for how they matter, we refer to our response to your *Line 280* comment.

*Table 3:* How were these listed number density variance values selected, and how do they affect the retrieval? As mentioned earlier, I'm particularly curious how the solution might be affected when the a-priori number density profile differs from the true atmosphere by a factor of 10 or more.

In regards to the covariance, we will again refer to our response to your *Line 280* comment. However, to your particular comment of a-priori profiles we have added content in the paper around a comparison with the SAGE III particle size data which address that question.

*Lines 405-406:* "However, of note this a-priori N profile is not constructed from any specific knowledge of the aerosol to be retrieved." What does this mean? I guess the a-priori profile is some kind of climatology, or ...?

We simply mean to say that this a-priori was built as a very simple climatology, and not with a-priori knowledge to the specifics of the observations - i.e. using reports from other instrumentation to build an idea of what to expect. The a-priori profile we use is just a file with steadying increasing aerosol from high to low altitudes, including an extra contribution around 20 km representing the Junge layer.
The text of the paper has been adjusted to more clearly reflect this.

*Figure 10:* This caption ends with the statement that "This small error primarily results from the horizontal averaging." For a small error, maybe I shouldn't quibble, but I have to ask: What quantity is being averaged horizontally, and how? Is this averaging discussed elsewhere in the paper, and is it obvious how you determined that this averaging is the cause of the observed error?

We believe we understand the confusion. The radiance profile is built by averaging columns of the images together. The averaging is along the horizontal dimension of the image - not in the vertical/altitude dimension of the image. Figure 5 makes note of this. In the case of the retrievals shown in the paper, we binned all 512 pixels on each row of the detector to get the radiance profile measurement vectors. This binning significantly reduces the noise versus using a single pixel of the row. What we are discussing in the caption is that the error bars of the measurement vector are plotted, they are just very small because of the averaging.

We have changed the caption of Figure 10 in hope of being more clear:
[...] Error bars of the measurement are shown, but too small to be easily visible. This relatively small error of the measurement vectors primarily results from the column binning of the ALI images to produce the radiance profiles.

*Figure 12:* The properties of the bimodal distribution are shown in the figure, but do you have a reference for how these particular properties were selected?

No reference in particular. To alter a size we just increased the radius with a correspond decrease in width. This is to approximate larger aerosols coagulating with smaller ones. Beyond that we simply made the two profiles with distinct particle sizes in altitude with respect to each other.

*Lines 459-460:* "This is a similar approach to the standard retrieval approach of OSIRIS and OMPS (Rieger et al., 2019; Taha et al., 2021)." For the latter case, the assumed aerosol size distribution is not log-normal. As stated at the start of the review, I appreciate the experiments with bimodal log-normal distributions to test the approach, but were any non-log-normal experiments also done?

No, non-log-normal distributions were not within the scope of the work we did.

*Lines 465:* "... yields respectable agreement overall." How is this defined? (With the extinction plots presented on a logarithmic scale, it's difficult to read the percentage error well.)

Added percent difference plots and quantified wording.

*Line 503:* "... atmospheric DoP can be well retrieved" – this should be quantified. quantified wording.

*Line 515:* "... but we this is a point" – should be "but this is a point." Corrected

*Line 519:* "... extinction in very good agreement" – this should be quantified. quantified wording.

*Line 521:* "... showed an overestimation of aerosol extinction" – this should be quantified. quantified wording.

*Line 503:* Is listing data as "available upon request" adequate? I understand that public release of the full set of ALI measurements may not be possible. But inclusion of the data used and illustrated in this study (as a public "supplement" file) seems like a reasonable expectation, which I thought had become a fairly standard practice (for data that is not "officially released" and archived elsewhere).

We will look into hosting ALI data on Zenodo (or something similar) in the very near future, and we will update the data availability statement when it is somewhere accessible.

---

## Author Comment (AC3)

**Author (Daniel Letros) responses to the referee comments is shown in blue.**

**We would like to offer our appreciation for your efforts and feedback. While some of your notes may be outside of what we consider the scope of the paper to be, we would like to say again that they are not bad suggestions. Much of what you suggest is of significant interest to us as well and it is simply a matter of time and resources to investigate them. Regardless, we feel your feedback has helped greatly to make this paper more presentable. Thank you!**

**Referee report to the "The Aerosol Limb Imager: Multi-spectral Polarimetric Observations of Stratospheric Aerosol" manuscript by Daniel Letros et al.**

The manuscript describes a retrieval algorithm for the new Aerosol Limb Imager (ALI) instrument and presents some results from the synthetic retrievals and from three example measurements made by the instrument. Although the measurement concept of ALI is similar to that used by the upcoming ALTIUS mission of ESA, it offers an unique feature of measuring the polarization state of the limb-scatter radiance. Authors did a great job showing how this feature of ALI can be used to detect contamination by clouds, which has always been an issue for limb-scatter aerosol retrievals. Less impressive are, however, the results from the aerosol retrieval itself. Here, I got an impression that the polarization-sensitive measurements make the instrument useless for the aerosol retrieval. This is not an issue for the scientific significance of the paper but the authors, especially PIs of the project, should think about if they really want to provide this impression to the scientific community. Unfortunately, the presentation of the results is elaborated quite poor and needs to be improved. To improve the message of the paper authors need to quantify the required conditions for the retrieval of reasonable aerosol extinction coefficient profiles from real measurements. A careful proof-read of the paper is needed to correct typos, extra or missing words etc.

There are two points we would like to generally highlight:

First, we understand there are some issues with the direct comparison of ALI retrievals against the results of SAGE III/ISS, OMPS, and OSIRIS. Principally, ALI retrieved extinction is biased high with respect to the other three instruments. We do not consider that such bias is unreasonable given our measurement platform of a high-altitude balloon (altitude of 36-37 km), which imposes analysis difficulties that are significantly less relevant to a satellite platform in orbit. We also do not think it invalidates the core of our demonstration.

However, we did fail in properly contextualizing and acknowledging this as the paper was. As a broad response to your feedback we have added more discussion (primarily around Fig 15 of the paper) which attempts to provide this context and acknowledgement. This also relates to some of your other feedback given below that we address further in those responses.

Secondly, there are a number of comments below that we would like to give a broad address to. The intended scope of this paper is to provide demonstration of the ALI instrument concept from a polarized limb measurement level (i.e. the Timmins 2022 campaign) through the current analysis algorithms we have developed arriving at a level 2 product of aerosol and cloud height. This demonstration culminates in the comparison of ALI results with SAGE III/ISS, OMPS, and OSIRIS for three example sets of ALI data. We provide simulated exercises to contextualize and support this comparison, and consider that these three scans well represent both the efficacy and limitations of our approach at the current state.

We feel that some of the comments below - while not bad suggests to give - tend to expand the already lengthy paper outside of what is required for this intended scope. We still respond to all comments below, but if we include an "out of scope" comment in our response we simply mean to say that we consider what is requested to be beyond the scope we have defined here.

**General comments**

- Date and time of the ALI measurements and those of the collocated reference measurements are not

provided. This make impossible to understand which aerosol conditions were investigated. Furthermore, no information about collocation criteria is given.

  – We have adjusted Table 1 to include more relevant information to the ALI retrievals rather than the flight gondola. Additionally, we have added a supplemental document which contains further information about the collection and coincidence of the ALI measurements to the other three instruments.

- Simulation results do not look representative enough. The aerosol extinction above 30 km was set to zero and no attempts were made to check what happens if a realistic aerosol distribution at high altitudes is included. It is not clear for which surface albedo the simulations are done and what happens if albedo changes. How the estimations of the surface albedo are affected by the presence of the aerosol above 30 km? Are the simulation results remain the same if an aerosol profile for different aerosol loading conditions is used? Logarithmic plots and absence of relative difference plots make evaluation of the retrieval quality nearly impossible. If I understand it correctly, the synthetic retrievals were done without adding measurement noise to the simulated spectra. How well the retrieval chain works if measurement noise is added (I mean here using noisy simulated spectra, not only adding the noise covariance matrix into the retrieval)? How the aerosol parameters for the exercise with bimodal aerosol PSD were set, where comes the information about the used parameters from? Are they realistic? Some quantification of the results in the case of the bimodal PSD is needed, i.e. some realistic bimodal distributions corresponding to different aerosol loading conditions have to be found and used for simulations. Comparison of the mode radius and width from the unimodal and bimodal distributions does not make much sense. A comparison of the effective radii might be more useful.

We will address different points here individually:

  – Simulation results do not look representative enough. The aerosol extinction above 30 km was set to zero and no attempts were made to check what happens if a realistic aerosol distribution at high altitudes is included.

  – How the estimations of the surface albedo are affected by the presence of the aerosol above 30 km?

    * We are unsure what you mean by a realistic aerosol distribution in this context. It is our understanding that it is realistic to consider sulphate aerosol zero at (approximately) 30 km and above. Of course there is some latitude dependence, but it is at about this altitude that the sulphate aerosol will evaporate into gas. As we also understand it, it is typical for Lidar aerosol measurements to use altitudes of 30 km and above as a Rayleigh signal for calibration. For the purposes of the paper, this assumption is supported by the observations of SAGE, OMPS, and OSIRIS which we include below. We do understand that it is not strictly correct assumption to make - **and this is an aspect of our address around Fig 15. in the paper which pertains to this and other limitations of our approach** - but this is an assumption we must make from a balloon platform which does not have access to even higher altitudes to normalize from.

[Figure]

High altitude aerosol extinction. Coincident profiles of Scan 1 used.

* To elaborate further on the normalization: when normalizing the measurement vectors, the aerosol one retrieves will be done with respect to the aerosol signal within the normalization altitudes. Again, ideally this aerosol signal would be zero and we make this assumption implicit in the paper due to limitations of the balloon platform. If this assumption is incorrect then the retrieved aerosol profile will be biased low by some proportion to the signal in the normalization.
* Robustly exploring this in simulation for different aerosol loading and/or albedo is computationally non-trivial, and we consider it out of scope of this paper given what we already present.

– It is not clear for which surface albedo the simulations are done and what happens if albedo changes.

* For the simulations, unless explicitly stated (i.e. the albedo estimation discussion), the albedo is held the same in both the simulated atmosphere of the true state and the forward model of the retrieval. This removes it as a factor in these simulations. In general, if the albedo between the true and retrieval atmospheres is incorrect then the aerosol will be biased. If the albedo is too low in the retrieval versus the true state atmosphere, then the missing scattered signal will be attributed to aerosol by the retrieval and the retrieved aerosol profile will be biased high. If the albedo is too high the opposite happens. Again, we consider robustly exploring this in simulation out of the paper's scope - **but it is a part of our new discussion around Fig 15. of the paper**.
* For the flight retrievals the albedo estimation is an effort meant to align the forward model with the up-welling signal seen by ALI. However, the high-altitude normalization is the main strategy to marginalize the effect of albedo.

– Are the simulation results remain the same if an aerosol profile for different aerosol loading conditions is used?

* Yes, the results of the simulation reflect well the performance of the retrieval even with different aerosol loading. The caveat to this statement is if true state conditions and a-priori conditions differ significantly - i.e. the underlying a-priori information given to the Kalman filter is no longer appropriate to reach the true state solution. This is a broad topic and a limitation of any Kalman filter applied to any problem (not just atmospheric inversions). As such the exercise of prototyping a Kalman filter and exploring it in simulation to yield appropriate performance is always necessary.
* We have added particle size comparison to SAGE III/ISS, and with that we also now discuss the influence of a-priori selection to a degree. However, presenting different aerosol loading in simulation (true state or the a-priori) is beyond the scope of this paper.

– Logarithmic plots and absence of relative difference plots make evaluation of the retrieval quality nearly impossible.

  ∗ We have added relative difference plots to the results now.

– If I understand it correctly, the synthetic retrievals were done without adding measurement noise to the simulated spectra. How well the retrieval chain works if measurement noise is added (I mean here using noisy simulated spectra, not only adding the noise covariance matrix into the retrieval)?

  ∗ We believe there is confusion here. The synthetic retrievals were done with measurement noise applied to the simulated observations constructing the measurement vectors, and not just in the noise covariance matrix. To avoid any further confusion we have now noted this explicitly at the beginning of Section 4.3.1.

– How the aerosol parameters for the exercise with bimodal aerosol PSD were set, where comes the information about the used parameters from? Are they realistic? Some quantification of the results in the case of the bimodal PSD is needed, i.e. some realistic bimodal distributions corresponding to different aerosol loading conditions have to be found and used for simulations.

  ∗ We do not have any particular reference for the bimodal aerosol distributions. We simply altered the sizes ourselves by increasing the radius with a correspond decrease in width. This is to approximate larger aerosols coagulating with smaller ones. Beyond that we simply made the two profiles with distinct particle sizes in altitude with respect to each other. Since these parameters do not vary significantly from the a-priori sizes we use (0.08 microns and 1.6 width which is also used by OMPS-LP) we consider them realistic.

  ∗ We agree that a robust study of the impact bimodal aerosol loading has on polarized unimodal aerosol retrievals (beyond what we show) would be very interesting. However this study will be *very* time intensive and computationally expensive. For now we consider it to be outside the scope of the paper.

– Comparison of the mode radius and width from the unimodal and bimodal distributions does not make much sense. A comparison of the effective radii might be more useful.

  ∗ This is a very good thought, and we show the effective radius of the bimodal simulations below. However, in the specific case of simulation (where the true state is known) we do not think that they add any new insight or information beyond what we already show.

[Figure]

(a) Exercise of Figure 12 (geometry of Scan 3)

[Figure]

(b) Exercise of Figure 13 (geometry of Scan 1)

Figure 1: Aerosol effective radius of bimodal simulations.

          * To elaborate more: a core finding we are attempting to convey in the paper is that polarized limb measurements are more sensitive to aerosol particle size distributions. Unimodal treatment of a bimodal distribution can be insufficient. To this end, failing to retrieve the correct extinction in the exercise of Figure 12 (using the geometry of Scan 3 where the atmospheric degree of polarization is higher than Scan 1) shows this shortcoming. Likewise, simulating the same aerosol state but in a less polarized atmosphere (geometry of Scan 1) in Figure 13 yields noticeable improvement. The effective radius profiles ultimately reflect the same improvement the extinction profiles do. We also attempt to convey that a similar looking improvement between Scan 1 and Scan 3 in the real retrievals (Fig 16 and Fig 18) is observed.

- Comparison with measurement data is difficult to evaluate without having the relative difference plots. It is not clear for which aerosol conditions the comparison with reference data was made, this information is extremely difficult to derive from the logarithmic plots. The overall conclusion from the comparison seems to be that only one of the three measurements produces reasonable results although data from both OSIRIS and OMPS-LP work well in all three cases. Is it a general issue of the applied technique? Is the technique then useful at all? A more detailed discussion of the usability of ALI results needs to be done at this point.

    - As we noted above, we have now added some relative difference plots for the extinction comparison. To better discusses the usability of ALI we have augmented the conclusion with the hope of addressing this concern.

**Detailed comments**

- Line 6 and 34: At this point it is unclear what the term "scalar width" means. Clarified.

- Introduction lacks some information about how the ALI instrument is ranged with respect to the previous, present and planned space-borne aerosol instruments.

    - The version of ALI in the paper is of course a balloon instrument, and while the ALI concept is planned to become a space-borne instrument, that version is still in development. Perhaps we misunderstand the intent of this comment, but we are unsure how to appropriately range ALI with respect to space-borne aerosol instruments beyond what we do for these reasons.

- Section 2: Some technical data of the instrument need to be provided, e.g. vertical/horizontal resolution and sampling, FOV range, typical exposure time etc.

    - We have added some more detail to the beginning of Section 2:
        * ALI is designed to image the atmospheric limb from a float altitude above 35 km. It has a 6° full field of view in the vertical dimension and a 4.8° in the horizontal. To image the limb with this field of view ALI is tilted down about 3° so the top lines of sight are flat and level with the (idealized) surface of the Earth. This external field of view is mapped to a $640 \times 512$ pixel detector which, including the instrument point spread function, yields a 0.06° angular resolution for each pixel in both dimensions of the external field of view. This translates to a tangent altitude resolution of $< 100$ m. Exposure times vary on ALI configuration (i.e. imaged wavelength) and the solar scattering geometry, but it is typically a few hundred milliseconds.

- Lines 51 - 52: "The SPS also contains one linear polarizer after the LCR and another after the AOTF to further refine the polarized image" – There is a polarizer before the AOTF and AOTF itself passes one polarization direction through. Please explain shortly why an additional refinement is needed.

    - We have added a short explanation: [...] the AOTF to further refine the polarized image (Letros et al., 2024; Kozun et al., 2021). The refinement provided by the polarizers heavily attenuates any unwanted light which may otherwise be passed through the AOTF.

- Figure 1 is quite dark and depending on the monitor and illumination difficult to see. Please light up the dark parts of the figure. We have brighten up the figure by making the black objects a lighter gray.

- Figure 2, panel (b): the abbreviation "DE" is not defined in the caption. Corrected.

- Line 69: "The spectral bandpass of ALI for a tuned frequency is taken as the full-width half-max (FWHM) of the diffraction response." – it looks like, here, you use the "diffraction response" term to describe the same function as shown in the panel (b) of Fig. 2 an referred to as the "spectral response". In line 75 you write, however "spectral response is calculated by integrating the measured diffraction responses" which suggest different meaning of these two terms. Please clarify.

  – Yes we consider the diffraction response of the AOTF and the spectral response of ALI to be the same thing. In line 75 we write "the area of each spectral response is calculated by integrating the measured diffraction responses". We use diffraction response here just to emphasize it is an AOTF measurement, but we understand it may be confusing.

  – We have changed line 75 to read: [...] the area of each spectral response is calculated by integrating the measured responses in Fig. 2. (b) [...]

- Figure 3 caption: "LCR on state" – here and throughout the text, it would be less confusing is you wrote "on" in the quotation marks. The same is for "off".

  – We have actually received the exact opposite feedback in the past from others. Our initial instinct was to write it as you suggest, but we think without quotations is also just fine. Suspecting we can not make everyone happy on this one, we will elect to keep it as currently written.

- Line 99: please give some details to explain what the "calibrated broadband integrating sphere of known (randomly polarized) spectrum" is.

  – We are not sure how to provide more detail which is both relevant and appropriate here to satisfy your comment. An integrating sphere is a common piece of optical equipment used in calibrations. We are simply saying we are using one which has been calibrated, and that we know well the spectral output of the source.

- Line 100: "Furthermore, since they reproduce the spatially flat and full-field conditions of the integrating sphere, they can be used to relate the ALI measurements to the external source each pixel measured."– Does "spatially flat" means homogeneous of a flat shape of the source? It is not quite clear how to get a response to an inhomogeneous source signal if you have a response to a homogeneous full-field signal. Please explain. Maybe I misinterpret what you want to say with this sentence, in this case please rephrase for more clarity.

  – Yes the integrating sphere is providing a flat/uniform source for ALI to image during calibrations. This sphere also illuminates all field angles of ALI just as the atmosphere does. Each pixel of the detector effectively just counts how many photons land on it, and since we know the intensity of the integrating sphere we can relate the raw detector images to units of radiance on a pixel by pixel basis. As for the response of an inhomogeneous source (like the atmosphere) that comes down to a matter of the instrument point spread function which defines our spatial resolution. We cannot spatially sample the atmosphere better than our point spread function allows, and within this resolution the atmosphere is effectively a homogeneous scene. More details about the relation of ALI image intensity to the atmospheric radiance can be found in (Letros et al., 2024) which includes polarized aspects of this.

- Line 128: Are there any investigations of the temporal stability of the obtained calibration parameters and possible dependence on the ambient temperature and pressure, strength of the illumination etc. ?

  – Temporally yes. Calibrations are performed at different instances over time. For example, we do calibration procedures within the lab both before and after events like a flight campaign. Not only does this tell us temporal stability on long time scales, but also if something happened to the instrument during flight. We find no concern in temporally stability.

– In regards to temperature, much design effort was given to monitoring and controlling the temperature of the instrument during operations. Particularly to the SPS. Calibrations are performed around nominal conditions which are well maintained during flight except for ascent conditions through the tropopause - and we do not attempt measurements in conditions like this. However given this approach, with the exception of the detector, temperature is not considered a variable in image processing

– The gain of our detector was explored as part of the conversion image calibration. Including adjusting the source brightness of the integrating sphere. From lab measurements the gain is very linear within our operating parameters (exposure times, AOTF configuration, etc). So we do not take any variation of detector response with respect to source strength into account.

– Pressure and other aspects of similar granularity being to go beyond the scope of the calibrations we did.

- Figure 3 caption: "White dots in this image indicate the bad pixels as determined by the dark correction." – I do not see any white dots in panel (a), what do the black dots mean?

  – We think you mean Figure 4 here (the figure showing the synthetic reproduction of an ALI image). There are white dots in Figure (a), but they are subtle. They are bad pixels as determined by dark correction only. The black dots in (a) which are white in (b) are pixels with poor response to illumination and are not yet flagged bad at the stage of processing seen in (a). We have adjusted the caption of this figure to be more clear.

- Figure 5 caption: The sentence "The radiance profiles of (b, c, e, f, h, i) are constructed by following Section 2.3 to convert images into photons/s/cm2 /sr from DN. Then following Section 2.1 to obtain units of photons/s/cm2 /sr/nm." should be moved to the main text. Agreed and done.

- Figure 5 caption: "the solar irradiance produced by a radiative transfer forward model" – solar irradiance is not produced by radiative transfer models.

  – Yes we can be more clear on the wording. We have changed "produced by a" to "used by the".

- Line 206: "a regularization matrix in place of $S_a^{-1}$ " – $S_a^{-1}$ is also a regularization matrix, do you mean Twomey-Tikhonov regularization matrix, i.e. smoothing constraint here?

  – We have clarified the wording to say we are not using a Twomey-Tikhonov regularization matrix.

- Line 208: "measurement and state vectors of large dynamic range which tends to produce ill-conditioned inversions." – the ill conditioning of the inverse problem is not caused by the large dynamic range of the measurement and state vectors. Its reasons are rather a dense vertical sampling, insensitivity of the retrieval to certain vertical ranges and correlation between different parameters.

  – We completely agree that the reasons you list can cause or contribute to ill-conditioning, but we think dynamic range can lead to ill-condition the retrieval too. For instance, constructing $\mathbf{S}_\epsilon$ as we describe in the paper (diagonal matrix of the measurement noise) with notable difference in magnitudes between wavelengths can ill-condition the $\mathbf{S}_\epsilon$ matrix. This leads to ill-condition of the gain matrix and averaging kernel.

- Line 214 - 217: In most previous aerosol retrievals the albedo and aerosol retrieval were run alternatively within an iterative process allowing the retrieved albedo to adjust to the retrieved aerosol profile and vise versa. In this retrieval, the albedo is retrieved only once before the aerosol retrieval. Additional investigations need to be done to show that the retrieved albedo does not depend significantly on the a priori aerosol profile used for the albedo retrieval.

  – We addressed the choice of this algorithm in the paper (first paragraph of Section 4.1). It is to remove albedo from the retrieval optimization and reduce the solution space the retrieval needed to consider. As for your call to additional investigation, we refer to our address in the general comments above. However, we will note again that our new discussion of our limitations (included around Fig. 15) touches upon the albedo.

- Line 221 - 222: I do not expect that the influence of aerosol at 33 – 34 km is minimal. This statement has to be proven using measurement data, e.g. GloSSAC.

  – We refer to our address in the general comments above.

- Sect. 4.1 please provide the aerosol profile used for albedo retrievals. Dependence of the albedo retrieval on the assumed aerosol profile needs to be investigated.

  – We refer to our address in the general comments above.

- Lines 245 - 246: "Therefore, we also present an approach to retrieve the Stokes parameters of the atmosphere in the Stokes basis of ALI ..." – Please explain what "the Stokes basis of ALI" means.

  – The Stokes basis is of course the coordinate frame in which the Stokes vector is being described in. By Stokes basis of ALI, we simply mean we retrieve the Stokes parameters as they look to the ALI instrument. If for instance the flight gondola is rolled, the Stokes basis of ALI will be different than one aligned to the Earth's horizon (or some other frame of reference).

- Eq. (7): What happened to U and V components. Were they just neglected? Please clarify.

  – Equation 5-7 note proper definitions, as well as approximations of $I$, $Q$, and the DoP $P$. The approximations can be built directly from ALI measurements (without the more sophisticated retrieval approach). So yes, $\tilde{P}(\lambda)$ in equation 7 ignores $U$ and $V$ in the approximation.

- Lines 267: "...which directly correspond to tangent altitudes at the time the observation was taken" – What do you want to highlight with "at the time the observation was taken", are the tangent heights measured not at the same time?

  – This sentence is related to equations 8 and 11 which indicate elements of the vectors by detector pixel. With changing orientation of the gondola, any particular pixel may measure a different tangent point and altitude over time. This sentence is indicating that this relation is known, and the retrieved value of the $n^{\text{th}}$ pixel can be related to altitude.

- Lines 300 - 302: "In this example $\Theta$ received little to no action by the inversion to adjust it form the a-priori state. It is sensible to simply not include $\Theta$ as a property in x and just rely on the a-priori values in the forward modeling." – Is it a common behavior or does it just happen occasionally?

  – Yes, in simulation we found this to be a very common behaviour.

- Line 310: "... we convolve the smoothed DoP with a central difference impulse response.." – please explain what is the "a central difference impulse response" or give a reference.

  – Here we are just describing taking a numerical derivative with the central difference method. We do this through a convolution, and in this vernacular it is called the impulse response.

- Fig. 7: Please provide an illustration what happens if there is a strong aerosol level with larger particles in place of the cloud layer.

  – Ultimately this is a point of future work for us, and we feel including this in the paper at this time will quickly lead out of the intended scope. Fig 7. depicts the cloud detection method of ALI in simulation. This, as presented in scope, encompass the retrieval of a quantifiable DoP used to identify altitudes of depolarization. The second paragraph of Section 4.2 states our intended scope and alludes to future interest.

  – To elaborate more: depolarization is associated with clouds because of enhanced scattering and anisotropy of the scatters (i.e. ice particles). In this work we model aerosols as isotropic sulphate droplets (which is a standard way to model stratospheric aerosol). We consider that the natural polarized distinction between the isotropic and anisotropic scattering gives a significant advantage to cloud detection. However, meaningful and robust investigation into the efficacy of cloud detection in the presence of larger aerosols will require modelling a slew of relative ice and aerosol

sizes/concentrations at different solar geometries/wavelengths (i.e. discussion of Fig 9) to study. We simply haven't devoted the resources to do this yet. That said, within the particle sizes we explored we haven't found ambiguity with the cloud properties we have explored.

- Line 333: "At each scan, the true state of the atmosphere includes GloSSAC aerosol but unlike the exercise of Fig. 7 and Fig. 8 no ice layer is included." – please show the aerosol profile used in the retrieval (if you like in the Supplement) and prove the results are the same for a different aerosol loading scenario.

  – The aerosol extinction we use is the same profile we show in Figure 10, but we don't consider the aerosol a primary driver of this effect. This manifests from the relative balance of horizontally and vertically polarized light within the atmosphere, which is largely driven by the scattering angle. The purpose behind the exercise is to demonstrate the failure of the DoP retrieval as a function of the ALI Mueller matrix. We believe the contrast between the top and bottom rows of this figure accomplish that as is.

- Sect. 4.2.4: It would be interesting to have a larger statistics about the working and non-working DoP retrievals.

  – We assume by larger statistics you mean more case studies (i.e. different solar geometries and wavelengths). We do agree and this is on-going work. For the paper though we consider this out of scope.

- Line 349: "retrieved along side a scalar width" – you still haven't properly defined what scalar width means.

  – Addressed in your first detailed comment.

- Line 369: "as the finest resolution A produced" – What does "A" mean here?

  – The averaging kernel **A**. It is bold in line 369 and defined in line 187. Although we see now the confusion it may cause. We have changed it to simply call it by name here.

- Lines 368 - 370: Resolution might be limited by the forward model grid but not determined by it. It is mainly determined by the instrument FOV and sampling and also applied regularization. What you mean here is the sampling.

  – We think we agree fundamentally, but perhaps there is confusion over wording. We are defining the altitude resolution in the forward model by determining the finest resolution which no longer increases the quality of the averaging kernel - i.e. if we increase the model grid resolution from 600 m to say 400 m, the resolution indicated by the averaging kernel will still be 600 m. In this case the measurement vectors we are using (for reasons like you note) lack the information to solve the state at a higher resolution. We have changed this sentence in the paper to try and avoid the confusion.

- Line 371: "we do not employ regularization" – Using Sa is also a regularization.

  – Addressed in comment of line 206.

- Lines 391 - 394: The description looks like the Levenberg-Marquard method, why do not you use the name, is the any significant difference?

  – Yes the dampening matrix **D** follows from the Levenberg-Marquard method. We have now included the explicit name when **D** is introduced.

- Sect. 4.3.1: Please show the GloSSAC aerosol profile used for the study. Why was only one profile used? Dependence on the true aerosol profile might be significant. Different aerosol loading conditions need to be investigated.

  – We refer to our address in the general comments above.

- Line 400: Please provide the illustration what happens if you do not set the aerosol profile above 30 km to zero.

  – We refer to our address in the general comments above.

- Figure 10: Is it correct that the true surface albedo is not changed here?

  – Yes, the albedo matches between the forward model and the true state.

- Line 416: What happens if a priori and true widths are the same but the width is included into the retrieval?

  – Good question. Using the a-priori covariance of 0.0001 for the scalar width tells the retrieval there is a 68.2% confidence (one sigma) that the true state width matches the the a-priori value width within 0.01. Because of this statistical freedom, the retrieval will end up moving the width off of the (correct) a-priori value. However, this movement is minimal and less than what we show in Figure 11. Of course one can tighten the covariance to reduce this effect, but these details are aspects of prototyping and tuning the retrieval algorithm.

- Figs. 10 - 13, 15, and 16 must include relative differences. Added.

- Line 433: "We wish to emphasize that the limb measurements of ALI are polarized, and we speculate that this polarized content contains useful information about the aerosol phase scattering matrices - which is of course influenced by the particle sizes" - also non-polarized measurements contain information on the particle sizes.

  – Yes, size information in general is available (although still challenging to retrieve) with measurements over wavelengths and/or scattering angles. Here we wish to emphasizes that the polarization information itself contains aerosol size information in addition to that obtained over wavelength and scatting angle.

- Sect. 4.3.1: Authors found a strong dependence of the results on the presence of the second mode for highly polarized measurements, the question is if polarized measurements are really an advantage or a drawback. This should be discussed in more details in the text.

  – We have attempted to make more clear the message we wish to deliver in the text. However to briefly clarify here as well: a purpose of the bimodal simulations was to showcase the failure of the retrieved measurement vectors reproducing the measurements under a higher degree of polarization. Similar behaviour is seen in the real retrievals as well. We would argue that this failure is a (at least potential) indication of the presence of more complex sizes - which itself is size information.

- Lines 448 - 451: "With respect to the simulations done in the exercise surrounding Fig. 9, we find almost all behavior regarding Fig. 14 to be expected including: the relative balance between horizontal and vertical polarizations for all three scans, the spectral regions expected to fail given the polarimetric response of ALI in Scan 1 and Scan 3, and the relative magnitude of the DoP for Scan 1 and Scan 2." – this discussion is difficult to follow, please write in more details what exactly you expect from the discussion around Fig. 9 and what we see in this respect in Fig. 14.

  – We have removed the reference to Fig 9. and it now reads "With respect to the simulations". We think this reference is where the confusion may arise. We are simply trying to say that polarized aspects of the atmosphere in all three scans are expected from forward modelled simulations - except the magnitude of the DoP seen in scan 3.

- Lines 448 - 451: "The simulation of this geometry done for Fig. 9 produced a true state DoP approximately ranging between 0.3 - 0.4" – please provide corresponding plots, if you prefer they can be in the Supplement.

– We have again removed the reference to Fig 9 to perhaps relieve confusion. We do not intend to include the plot of the DoP you request in the paper or supplement as we believe it adds nothing of value above what we state in text - but we do show it here immediately below. The significant difference we find between Fig 14 (c) and this plot is the magnitude of the DoP. We expect it to be in the range of 0.3-0.4 as the plot here shows.

– As Fig 7 and the Fig 14 (a) and (b) show, cloud (which is not in the forward modelled atmosphere for the DoP below) should yield DoP much lower then 0.3, and aerosol loading (which is included in the forward modelled atmosphere for the DoP below) should range roughly in the 0.3-0.4 range we see. The lump at about 18 km you see below comes from the aerosol loading (see the true state aerosol of Figure 10).

– As we say in the paper, we have no reason to consider the result of Fig 14 (c) to be wrong. We do not know what is yielding it yet. We suspect as the future work of exploring the DoP under different conditions (my response to your comment of Fig. 7. above) may yield more insight.

[Figure]

Figure 2: True forward modelled DoP of Scan 3 (exercise related to Fig 9)

- Line 473: "... of interest is the profile of r which indicates a layer of larger particles at approximately 22.5 km..." – please compare this results with SAGE III data and make a statement about the agreement. We have added a comparison to SAGE III particle size.

- Lines 485 - 488: A large disagreement for ALI and a good agreement for other instruments means, in my opinion, that the instrument concept where the polarized measurements are used is not optimal for the retrieval of the aerosol extinction coefficient. Please discuss this issue.

  – We acknowledge that ALI is an outlier to the other instruments. Our new discussion of limitations around Fig. 15 of the paper now addresses this more directly. However, the best extinction coefficient is built with accurate particle size. With what we have shown in the paper (and as you noted in the comment of Sect. 4.3.1 above), we have reason so think that the polarization includes useful size information which ALI can probe. Under our current unimodal approach, this has limiting factors which may (partially) explain why we are the outlier, but this is a point of ongoing work for improvement. We have attempted to convey this message better in the overall text as well.

**Technical corrections**

All technical corrections below are accepted and changed in the paper. Thank you for your diligence in spotting them!

- Line 33 and throughout the text: "dependant" → "dependent"

- Figure 1 caption: "comprises of one off-axis" → "comprises one off-axis"

- Figure 2 caption: "shown in black" → "is shown in black"

- Figure 3 caption: "response of ALI (Letros et al., 2024) shown as" → "response of ALI (Letros et al., 2024) is shown as"

- Line 117: "be used to constructed" → " be used to construct"

- Line 128: "conversion to from DN into" → "conversion from DN into"

- Line 170: "Here we discuses the concepts" → "Here we discus the concepts"

- Line 175: "and marks the lower altitude limit to retrieve" → "and mark the lower altitude limit to retrieve"

- Line 347: "to emphasise that the in the context" → "to emphasise that in the context"

- Line 347: "criteria we set is to retrieve" → "criterion we set is to retrieve"

- Line 349: "their is" → "There is"

- Line 364: "ceiling on of the radiance" → "ceiling on the radiance"

- Line 396: "summary of retrieval result" → "summary of retrieval results"

- Line 407: "by retrieving a N and r profile" → "by retrieving N and r profiles"

- Line 413: "by the biased caused" - do you mean "by the biases caused"?

- Line 417: "shows the behaviour" → "show the behaviour"

- Line 515: "but we this is a point" → "but this is a point"

---

## Referee Report (RR1)

Referee report to the revised version of the "The Aerosol Limb Imager: Multispectral Polarimetric Observations of Stratospheric Aerosol" manuscript by Daniel Letros et al.

In their response the authors claim that additional investigations required by me to justify the validity of the presented approach are beyond the scope of their paper. Unfortunately, I cannot agree that the scope of the paper defined by authors is sufficient for the publication. The authors neither investigated most crucial error sources in the synthetic retrievals nor provided a statistically significant validation. At least one of these issues needs to be addressed properly to justify the trustability of the retrieval results. Although this information is not provided in the paper, I assume that one balloon flight does not provide sufficient number of measurements to do a statistically significant validation. Concerning the synthetic retrievals the following investigations have to be done in addition:

- 1. Investigate influence of the assumption about the negligible amount of stratospheric aerosols above 30 km. This must be done for both aerosol and albedo retrieval. In general, the statement made by the authors that this amount is negligibly small is false. Multiple studies report that the aerosol extinction around 30 km is about  $10^{-5}$  km-1 and is still above  $10^{-6}$  km-1 in the 32 33 km altitude range [1, 2, 3]. Thus, it is definitely non-zero in the 30 33 km range used by the authors for the normalization. It is essential to check how this amount of aerosols affects the retrieval.
- 2. Investigate how a change in the surface albedo affects the retrieval. In the current version of their retrieval, the authors do no change the value of the surface albedo when doing synthetic retrievals. This makes impossible to evaluate how the retrieved albedo value depends on the aerosol state and how the uncertainties in the albedo retrieval affect the resulting aerosol extinction coefficient profile.

**Detailed comments**

- Author replies, page 14: "The caveat to this statement is if true state conditions and a-priori conditions differ significantly i.e. the underlying a-priori information given to the Kalman filter is no longer appropriate to reach the true state solution."

  This formulation is too imprecise. How different the aerosol state should be to cause a retrieval failure? Is it still within realistic values? Will the retrieval fail, e.g. after a strong volcanic eruption like the one of Hunga Tonga?
- Author replies, page 19, my comment to Fig.7: The paper claims that DoP is a suitable measure to distinguish between the aerosol and clouds. To justify this, it is not enough to show you can see a cloud, you also need to show that there is no false identification for strong aerosol levels. Furthermore, it is unclear if the method also works for water clouds.

- Introduction: I do not understand why you cannot describe how the capabilities of the ALI balloon-borne instrument are compared with those of current satellite instruments.
- Page 2, line 55 (tracked changes version): ".... so the top lines of sight are flat and level with the (idealized) surface of the Earth." → I do not understand what you want to say here, please reword the sentence.
- Fig. 1 caption: please spell out SPS and LCR also in the figure caption.
- Page 6, line 119 (tracked changes version): please replace "flat" by "homogeneous".
- Page 6, line 136 (tracked changes version): please replace "flat uniform" by "homogeneous".
- Fig. 1 caption: black dots in the left panel are still not explained.
- Page 7, line 143 (tracked changes version): please replace "flat" by "homogeneous".
- Page 7, line 172 (tracked changes version): "However, for the scope of the present work we select only three of these science scans for demonstration." → please write how many states are available in total and justify why only 3 of them are used.
- Page 15, line 340 (tracked changes version): "This albedo informs  $\theta$  of a simple Rayleigh atmosphere (no aerosol or ice) which constructs the a-priori profile."  $\longrightarrow$  I do not understand what you want to say with this sentence, please reword.
- Page 25, line 501 (tracked changes version): "produced a true state DoP approximately ranging between 0.3 0.4" → As you do not show the corresponding plot it is completely unclear where these values come from.
- Page 26, line 520 (tracked changes version): "This does still present a polarized aspect
  to the ALI retrievals of this exercise which hampers a completely level methodology to
  the comparison." → I do not understand what you want to say with this sentence.
  Please reword.
- Page 29, line 597 (tracked changes version): "we find encouraging indication of size agreement between ALI and SAGE III/ISS with the metric of effective radius."

  I think this statement is too optimistic.

• Figs. 17 and 18: Last sentences in the figure captions should be moved to the main text.

**Technical corrections**

- Page 15, line 342 (tracked changes version): "Fig. 7 is discussed"  $\longrightarrow$  "Fig. 7 are discussed"
- Page 15, line 348 (tracked changes version): "that is not"  $\longrightarrow$  "that are not"
- Page 20, line 329 (tracked changes version): "consists of the 10 wavelengths"
   → "consists of the 10 wavelengths"
- Page 21, line 460 (tracked changes version): "states is well" \rightarrow "states is well"
- $\bullet$  Page 28, line 587 (tracked changes version): "is shown"  $\longrightarrow$  "are shown"

**References**

Rieger, L. A., Zawada, D. J., Bourassa, A. E., Degenstein, D. A., A Multiwavelength Retrieval Approach for Improved OSIRIS Aerosol Extinction Retrievals, J. Geophys. Res. Atmos., 124(13),7286-7307 https://doi.org/10.1029/2018JD029897, 2019.

Pohl, C., Wrana, F., Rozanov, A., Deshler, T., Malinina, E., von Savigny, C., Rieger, L. A., Bourassa, A. E., and Burrows, J. P.: Stratospheric aerosol characteristics from SCIAMACHY limb observations: two-parameter retrieval, Atmos. Meas. Tech., 17, 4153-4181, https://doi.org/10.5194/amt-17-4153-2024, 2024.

Kovilakam, M., Thomason, L. W., Verkerk, M., Aubry, T., and Knepp, T. N.: OMPS-LP aerosol extinction coefficients and their applicability in GloSSAC, Atmos. Chem. Phys., 25, 535-553, https://doi.org/10.5194/acp-25-535-2025, 2025.

---

## Author Response (AR2)

Author (Daniel Letros) responses to the editor feedback is shown in blue We would again like to extend our sincere thanks to to all referees and the editor for their efforts.

Thank you!

**Editor Comments:**

The authors should check their final manuscript carefully for language errors. For example, at some places "affect" should be "effect", "employees" should be "employs", etc..

Effort was again made to ensure correct language.

Please ensure that the colour schemes used in your maps and charts allow readers with colour vision deficiencies to correctly interpret your findings. Please check your figures using the Coblis – Color Blindness Simulator (https://www.color-blindness.com/coblis-color-blindness-simulator/) and revise the colour schemes accordingly.  $\rightarrow$ Figs. 3, 9, 12, 15, 16, 17, 18

We have checked the plots and made color changes in the past to help address this. We admit they are still not ideal as we rely heavily on colors to convey information in the plots, but if they need further improvement we require advice to optimize them for all readers.

Please add the proper numbering to your supplement material (see "Supplements" at https://www.atmospheric-measurement-techniques.net/submission.html# assets)

Corrected.

**Referee #1 - Report 3**

line 14: "...with respected to..."  $\rightarrow$  "...with respect to..."

Corrected.

line 94: "...to account for the nm dependence..."  $\rightarrow$  "...to account for the wavelength dependence..."?

Corrected.

line 169: "...off imaging providing..."  $\rightarrow$  "...off imaging provides..."?

Corrected.

Figure 5 caption: why are there periods scattered through the caption, always before parentheses? Please revise.

Corrected.

Figure 5: This is a general comment about all figures: when creating sub-paneled plots, can the axes have the same scale, range, tick locations, and tick labels? That makes interpretation easier for the reader. This applies to all figures, but Fig. 5 provides a good example. e.g., the y-axis in (a) extends from an unknown altitude (0 km?) to 36.3 km, while (b) extends from another unknown altitude to something above 37 km. The reader should have to waste time trying to mentally align wonky axes, instead he should spend it understanding the science. Just a suggestion, but from a reader's perspective this would be very nice to have.

We see your point. Of course, the root issue is the same pixel in an ALI image will see a different tangent altitude as the altitude/attitude of the suspended balloon gondola changes. This is why the scale between (a), (d), and (g) in this figure are different (notably at the high altitude). Since in Figure 5 we are showing the whole images, we have opted to keep these figures as they are with the y-ticks noting (as well as possible) the common tangent altitude locations with respect to image pixel.

However, we have adjusted the y-scales of Figures 9 and 14 to be consistent with your feedback. These are two plots where we see the issue you raised exists without reason.

line 424: "...consists of the of the 10..."  $\rightarrow$  "consists of the 10..."

Corrected.

line 454: "...states is is well..."  $\rightarrow$  "...states is well..."

Corrected.

line 474: "...and we speculate..." No need to speculate! This can be demonstrated with simple Mie theory. Please consider rewording.

Reworded - removed statement of speculation.

Figure 14 (and subsequent): another example of mismatching tick locations, labels, and scales.

Y-axis have been adjusted and plot rescaled (Figure 9 as well).

**Referee #2 - Report 2**

I remain confused about the definition of the albedo. The response to the earlier comment (referencing line 226 in the earlier draft) states that the "true state" albedo is "... meant to indicate the known surface albedo of the simulated atmosphere (the atmosphere we are estimating the albedo of)." Are we discussing the albedo of a surface, or a surface/atmosphere system, or ...?"

In the simulation of the albedo estimation, this *effective* albedo is just the surface albedo because that is all that was simulated in this context. However, for the real measurements the *effective* albedo is not only the unknown surface albedo, but also the unknown atmospheric contribution to the up-welling radiation beneath the retrieval altitudes (i.e. clouds).

The authors pledge to "look into" hosting the ALI data in a public archive. For the small amount of data illustrated in this study, I still think that providing that ALI data as a supplement to this paper would be better, but I'll defer to the editors about the journal's requirements.

We have uploaded ALI data here: https://doi.org/10.5281/zenodo.15707122. This was indicated in the data availability section of our last submission.

**Referee #3 - Report 1**

Following the opinion of the editor, we are only addressing the comments and technical corrections.

Author replies, page 14: "The caveat to this statement is if true state conditions and a-priori conditions differ significantly - i.e. the underlying a-priori information given to the Kalman filter is no longer appropriate to reach the true state solution."  $\rightarrow$  This formulation is too imprecise. How different the aerosol state should be to cause a retrieval failure? Is it still within realistic values? Will the retrieval fail, e.g. after a strong volcanic eruption like the one of Hunga Tonga?

This statement in our response to your last feedback was intended to generally note that Kalman Filtering (like any optimization method) has limits. The importance of the a-pirori is well known from literature, but the specifics always depend on a particular application of the Kalman filter. This is why prototyping and tuning a filter with simulation is an important and necessary step. The simulation work we present in the paper is our demonstration of the efficacy and limitations our approach. Much of this relates the a-priori assumptions to the aerosol state and limitations of correctly retrieving the aerosol.

Author replies, page 19, my comment to Fig.7: The paper claims that DoP is a suitable measure to distinguish between the aerosol and clouds. To justify this, it is not enough to show you can see a cloud, you also need to show that there is no false identification for strong aerosol levels. Furthermore, it is unclear if the method also works for water clouds.

As we discussed in the response you are referencing, we agree there is more work to do on this aspect. While important and interesting, we are aligned with the suggestion of the editor and are allocating this to future work.

Introduction: I do not understand why you cannot describe how the capabilities of the ALI balloon-borne instrument are compared with those of current satellite instruments.

We have added a sentence in the introduction addressing a core difference between satellite and balloon limb instruments.

Page 2, line 55 (tracked changes version): ".... so the top lines of sight are flat and level with the (idealized) surface of the Earth."  $\rightarrow$  I do not understand what you want to say here, please reword the sentence.

We have reworded this sentence.

Fig. 1 caption: please spell out SPS and LCR also in the figure caption Done.

Page 6, line 119 (tracked changes version): please replace "flat" by "homogeneous".

Done.

Page 6, line 136 (tracked changes version): please replace "flat uniform" by "homogeneous".

Done.

**Fig. 1 caption: black dots in the left panel are still not explained.**

We believe you mean Fig.4. What we think you mean by black dots in the left panel (Fig.4(a)) are just part of the actual detector image, and deep blue on the DN color scale. These are dead pixels with poor illumination response. These pixels turn white in Fig.4(b) as this image is synthetically reproduced from the calibration coefficients, which has identified these pixels as non-ideal for discard. We already took action in the last set of revisions to note the different level of image correction between (a) and (b) in the caption of Fig.4.

Page 7, line 143 (tracked changes version): please replace "flat" by "homogeneous".

Done.

Page 7, line 172 (tracked changes version): "However, for the scope of the present work we select only three of these science scans for demonstration."  $\rightarrow$  please write how many states are available in total and justify why only 3 of them are used.

Detail added.

Page 12, line 253 (tracked changes version): "The measure we take to quantify the albedo is the integration of the high-altitude radiance with respect to wavelength"  $\rightarrow$  What happens if the surface albedo is wavelength dependent? Please check the wavelength dependence of the surface albedo for common surface types and investigate implications for the retrieval if albedo of any of them shows significant wavelength dependence in the relevant spectral range.

We would like to say again that we think this (and other suggested studies) are very good and interesting exercises to do. However, following from previous discussion, we are aligned with the suggestion of the editor and are allocating this to future work

Page 15, line 340 (tracked changes version): "This albedo informs  $\theta$  of a simple Rayleigh atmosphere (no aerosol or ice) which constructs the a-priori profile."  $\rightarrow$  I do not understand what you want to say with this sentence, please reword. Reworded.

Page 25, line 501 (tracked changes version): "produced a true state DoP approximately ranging between 0.3 - 0.4"  $\rightarrow$  As you do not show the corresponding plot it is completely unclear where these values come from.

We state that these values come from a simulation of this geometry with GloSSAC aerosol loading. We also provided the plot in our previous response on this topic. We still do not think the inclusion of the plot within the paper adds substance which the statement in text does not.

Page 26, line 520 (tracked changes version): "This does still present a polarized aspect to the ALI retrievals of this exercise which hampers a completely level methodology to the comparison."  $\rightarrow$  I do not understand what you want to say with this sentence. Please reword.

Reworded.

Page 29, line 597 (tracked changes version): "we find encouraging indication of size agreement between ALI and SAGE III/ISS with the metric of effective radius."  $\rightarrow$  I think this statement is too optimistic.

We are unsure of what action (if any) you would like us to take here.

Figs. 17 and 18: Last sentences in the figure captions should be moved to the main text.

Done.

Page 15, line 342 (tracked changes version): "Fig. 7 is discussed"  $\rightarrow$  "Fig. 7 are discussed"

Corrected.

Page 15, line 348 (tracked changes version): "that is not"  $\rightarrow$  "that are not" Corrected.

Page 20, line 329 (tracked changes version): "consists of the 10 wavelengths"  $\rightarrow$  "consists of the 10 wavelengths"

Corrected.

Page 21, line 460 (tracked changes version): "states is is well"  $\rightarrow$  "states is well" Corrected.

Page 28, line 587 (tracked changes version): "is shown"  $\rightarrow$  "are shown" Corrected.